# Immunogenicity and efficacy of CNA25 as a potential whole-cell vaccine against systemic candidiasis

Satya Ranjan Sahu[1,2,3], Abinash Dutta[1,3], Doureradjou Peroumal[1], Premlata Kumari[1,2], Bhabasha Gyanadeep Utakalaja[1,2], Shraddheya Kumar Patel[1,2] & Narottam Acharya [ID][1✉]

## Abstract

Disseminated fungal infections account for ~1.5 million deaths per year worldwide, and mortality may increase further due to a rise in the number of immunocompromised individuals and drug-resistance fungal species. Since an approved antifungal vaccine is yet to be available, this study explored the immunogenicity and vaccine efficacy of a DNA polymerase mutant strain of *Candida albicans*. CNA25 is a *pol32ΔΔ* strain that exhibits growth defects and does not cause systemic candidiasis in mice. Immunized mice with live CNA25 were fully protected against *C. albicans* and *C. parapsilosis* but partially against *C. tropicalis* and *C. glabrata* infections. CNA25 induced steady expression of TLR2 and Dectin-1 receptors leading to a faster recognition and clearance by the immune system associated with the activation of protective immune responses mostly mediated by neutrophils, macrophages, NK cells, B cells, and CD4+ and CD8+ T cells. Molecular blockade of Dectin-1, IL-17, IFNγ, and TNFα abolished resistance to reinfection. Altogether, this study suggested that CNA25 collectively activates innate, adaptive, and trained immunity to be a promising live whole-cell vaccine against systemic candidiasis.

**Keywords** Antifungal Vaccine; Cytokines; Neutrophils; Dectin-1; Trained Immunity
**Subject Categories** Immunology; Microbiology, Virology & Host Pathogen Interaction

## Introduction

Fungal pathogens are a serious concern to human health as they systemically disseminate and colonize various essential organs to cause sepsis and death (Brown et al, 2012; Sahu et al, 2022). About billions of people get infected and ~1.5 million of those succumb annually to fungal infections despite existing treatment modalities. Recently, WHO reported a first-ever list of fungal "priority pathogens" including a catalog of 19 fungi mostly belonging to the genus *Candida*, *Cryptococcus*, and *Aspergillus*, and emphasized rapid diagnostics, monitoring antifungal resistance, and development of novel antifungal drugs and immunotherapeutics (Fisher and Denning, 2023). Among *Candida* species, *Candida albicans* is the most common causative agent of invasive candidiasis globally, followed by other non-albicans species (NAC) like *C. glabrata*, *C. parapsilosis*, *C. tropicalis*, *C. dubliniensis*, *C. krusei*, *C. lusitaniae*, and *C. auris* (Brown et al, 2012). As a typical member of the human microbiota, *C. albicans* evolves as a harmless commensal in healthy individuals and maintains a long-term mutualistic relationship with the host by regulating microbial dynamics, metabolism, and immunity (Peroumal et al, 2022); however, its excessive growth causes life-threatening infections in immunocompromised persons with a mortality rate exceeding 40% (Brown et al, 2012). As these pathogens are becoming increasingly common and developing resistance to available antifungal drugs, the development of prophylactic and fungal vaccines becomes a priority. To date, no approved vaccines are available for human use, although studies on host–fungal interactions have identified several vaccine candidates and suggested those to generate varied degrees of protective immunity and were shown to be safe and effective against *Candida* and other fungal infections in preclinical models (Sahu et al, 2022; Tso et al, 2018).

Since whole-cell vaccines are highly immunogenic, both live and killed forms of such vaccines are effectively used against viral and bacterial infections. Heat-killed strains of *C. albicans* and *Saccharomyces cerevisiae* were tested to be partially protecting against experimental vaginal candidiasis (Cardenas-Freytag et al, 2002; Liu et al, 2012). A live whole-cell vaccine is a genetically altered whole organism that has just been attenuated such that it elicits enough protective immune response but does not cause disease in healthy people (Pollard and Bijker, 2021). Some of the genetically engineered *C. albicans* strains (tet-NRG1, PCA-2, CM1613, etc.) were reported to be attenuated and protective against disseminated candidiasis in mice to different efficacies (Sahu et al, 2022; Tso et al, 2018). However, further trials of these strains are yet to be initiated or disclosed. Since DNA polymerases are essential for the survival and replication of a pathogen, strains of *C. albicans* defective in DNA polymerase function could be attenuated and can generate protective immune responses to prevent both mucosal and systemic candidiasis. In eukaryotes, DNA replication is carried out by a coordinated function of three essential DNA polymerases Polα, Polδ, and Polε (Acharya et al,

[1]Department of Infectious Disease Biology, Institute of Life Sciences, Bhubaneswar, Odisha 751023, India. [2]Regional Center for Biotechnology, Faridabad, Haryana 751021, India. [3]These authors contributed equally: Satya Ranjan Sahu, Abinash Dutta. ✉E-mail: narottam_acharya@ils.res.in

2020). Extensive genetic and biochemical studies in *S. cerevisiae* suggested that Polδ plays a vital role in synthesizing both leading and lagging strands of DNA, during DNA repair synthesis, and recombination process (Acharya et al, 2011; Guilliam and Yeeles, 2020). While Polα-primase provides the RNA-DNA primer during the initiation of DNA replication, Polε can carry out leading strand synthesis (Burgers and Kunkel, 2017). In *S. cerevisiae* and *C. albicans*, the Polδ holoenzyme consists of Pol3, Pol31, and Pol32 subunits (Acharya et al, 2011; Patel et al, 2023). Pol32 subunit is dispensable in both the yeasts, however, in its absence, the processivity and fidelity of Polδ get compromised and *C. albicans* cells exhibit slow growth phenotype, sensitivity to DNA damaging agents, increased rate of loss of heterozygosity, and accumulation of indels and SNPs in the genome. Interestingly, animals infected intravenously with the *pol32ΔΔ* strain (CNA25) survived and did not develop any sign of systemic candidiasis (Patel et al, 2023). Therefore, in this study, we explored the immunogenicity and vaccine efficacy of CNA25 against systemic candidiasis in preclinical models. We found that upon immunization with CNA25, it elicits robust immune responses, and both immune-competent and -compromised animals were protected from the lethal challenges of various *Candida* species.

# Results

## Mice immunized with CNA25 exhibited protection against the pathogenic challenge of *C. albicans* and non-albicans species

Hematogenously disseminated candidiasis is caused by both *C. albicans* and non-albicans species. In our earlier study, we showed that unlike the WT strain that caused multi-organ sepsis to kill the animals, the systemic challenge of CNA25 did not induce any noticeable infection, and mice survived (Patel et al, 2023). To strengthen our result, now we compared the virulence of CNA25 with NAC species (Fig. 1A). While upon intravenous (IV) challenge with wild-type *C. albicans* and NAC species, mice ($n = 8$ per group) succumbed to infection ($P = 0.0003–0.0004$), CNA25-infected mice survived similarly to the saline control group (light blue and green lines, $P > 0.9999$). Notably, we observed a varied degree of virulence among *Candida* species when the survivability of the host was measured as a solo parameter. With the same inoculum size ($5 \times 10^5$ CFU) and within 30 days of infection period, *C. albicans* killed 100% of the animals within 10 days of infection (median death time (MDT) = 6 days), whereas the non-albicans strains did not cause any animal death (Fig. EV1A). However, a higher inoculum size (~$6 \times 10^6$ to $10^7$ cells) of NAC species induced animal killing with a significant delay in death time. The MDT was 9 days for *C. tropicalis* and *C. glabrata*, while that of *C. parapsilosis* was 20 days (Table EV1). To confirm the disease severity and death of the infected mice due to fungal infection, fungal load in the kidney tissues of euthanized mice was checked by CFU and PAS staining analyses (Fig. 1B). CFU analyses revealed that mice with high disease severity showed the maximum *C. albicans* load (~$3.5 \times 10^5$ cells/kidney) than the NAC species (~$2 \times 10^5$ cells per kidney) and a higher fungal burden seems to be associated with early animal death in *C. albicans* challenged mice (Fig. 1Bi). PAS staining of kidneys also confirmed the presence of respective fungal

species. While hyphal cells were predominantly observed in *C. albicans* and *C. tropicalis* infected kidneys, round but grouped cells were largely seen in *C. parapsilosis* and *C. glabrata* infected renal organs (Fig. 1Bii). This result suggested that *POL32* deletion of *C. albicans* suppressed the virulence attributes to become completely non-pathogenic and hence, we hypothesized that the animals immunized with CNA25 could be protected from the lethal challenges of *Candida* species. To validate it, groups of BALB/c mice ($n = 8$) were immunized intravenously with $5 \times 10^5$ CNA25 cells or 100 μl of saline per mouse as a control group, and after 30 days, they were re-challenged with *C. albicans* ($5 \times 10^5$ cells per mouse), *C. tropicalis* ($6 \times 10^6$ cells per mouse), *C. parapsilosis* ($6 \times 10^6$ cells per mouse), and *C. glabrata* ($1 \times 10^7$ cells per mouse) and their survivability was monitored (Fig. 1C). Interestingly, while the sham immunized mice succumbed to infections by *C. albicans* within 9 days ($P < 0.0001$), all the CNA25-immunized mice (100%) were protected (orange vs. green lines). More importantly, we observed a diverse degree of protection in vaccinated mice against NAC species. The CNA25-immunized mice showed complete resistance to *C. parapsilosis* ($P = 0.2482$), whereas partial protection was observed in *C. tropicalis* (~38% survived, MDT = 19 days, $P = 0.0088$) and *C. glabrata* (75% survived, $P < 0.9999$) re-challenged animals. This result suggested that CNA25 being an avirulent strain probably generates a robust protective immune response in the preclinical models to prevent fungal reinfection. All the protected mice that survived beyond 30 days in these and subsequent experiments were allowed to survive and kept on observation for another 6 months and then handed over to the animal care center for appropriate disposal.

Next, to find out the best possible route of effective immunization, we infected the mice groups ($n = 8$) with live CNA25 ($5 \times 10^5$ cells) by subcutaneous (SC), intraperitoneal (IP), and oral gavage (OG). Heat-killed (HK) CNA25 cells ($5 \times 10^5$ cells) were also injected into another group of animals intravenously. While one dose of inoculum was given for IV, SC, and IP routes, oral gavaging was carried out continuously for 7 days. After 30 days of primary inoculation, these mice were subjected to IV re-challenged with a lethal dose of *C. albicans* ($5 \times 10^5$ cells), and the survivability was monitored (Fig. 1D). While the HK CNA25 (IV) did not provide any protection to animals (Fig. 1C, black line, $P < 0.0001$), oral mode of CNA25 immunization only partially protected (~38%, $P = 0.3689$) but delayed the death significantly (Fig. 1D, brick red line). The median death time of 1°(oral) CNA25-2°WT was 22 days. The subcutaneous and intraperitoneal modes of vaccination only improved the survival duration (median death time = 8–10 days). These results suggested that live whole cells of CNA25 provided efficient protection through intravenous immunization. Henceforth, only the IV mode of immunization was followed for subsequent analyses.

## In vivo characterization of host–fungal interactions of CNA25 strain

*C. albicans* through systemic circulation reaches to the vital organs of the host where it colonizes to induce fungal sepsis and septic shock. To achieve colonization, *C. albicans* has to replicate in the niche site. Since CNA25 was avirulent, it was intriguing to determine the in vivo propagation status of CNA25 upon systemic inoculation. Three groups of BALB/c mice were intravenously

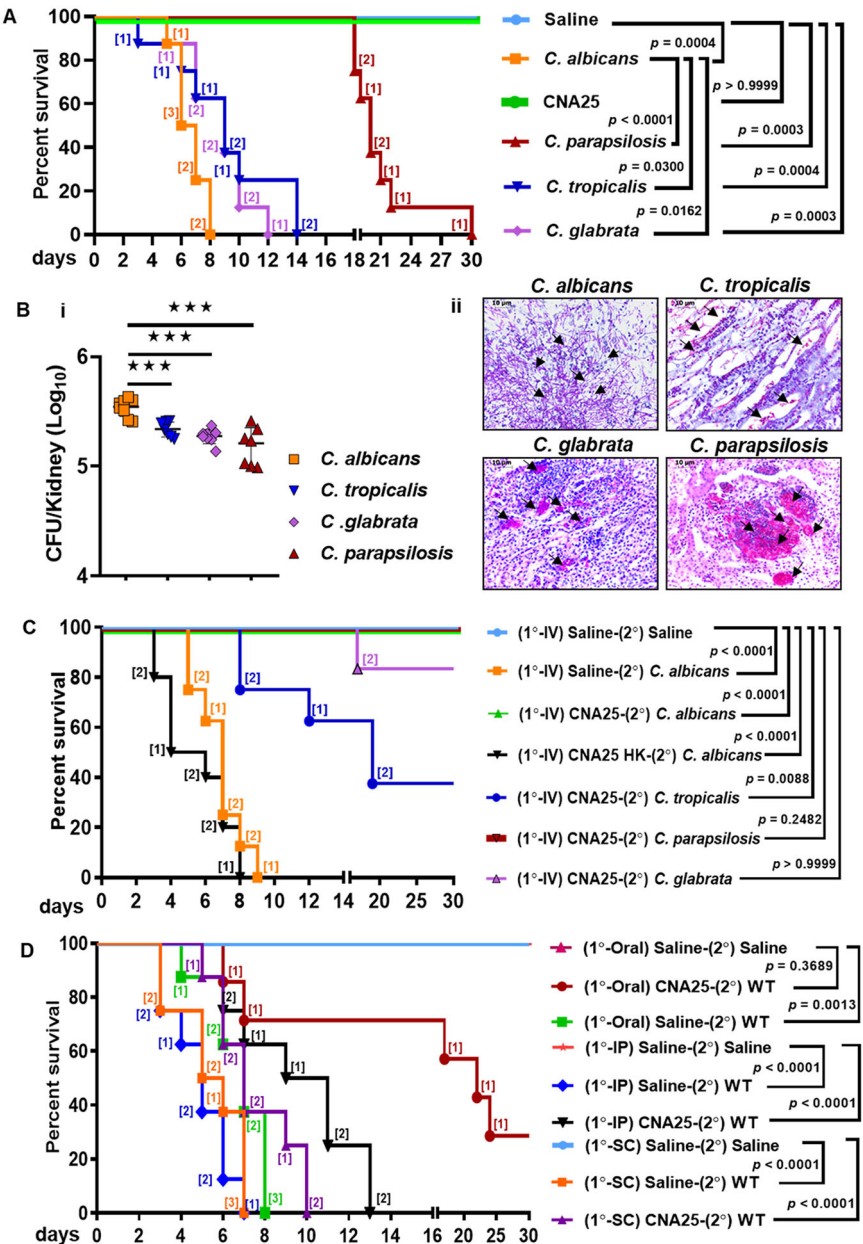

**Figure 1. Survival and protection assay with *C. albicans* and non-albicans species.**

(A) Survival analysis of BALB/c mice ($n = 8$) upon intravenous administration of CNA25 ($5 \times 10^5$ CFU/mice), *C. parapsilosis* ($6 \times 10^6$ CFU/mice), *C. tropicalis* ($6 \times 10^6$ CFU/mice) and *C. glabrata* ($1 \times 10^7$ CFU/mice) along with WT *C. albicans* ($5 \times 10^5$ CFU/mice) and saline control (100 µl 1× PBS) was carried out for 30 days and Kaplan–Meier survival curve was plotted. (B) Fungal burden in the kidney of mice that succumbed to various fungal-specific infections was determined by CFU analysis (i) and PAS staining (ii). Data were from eight biological replicates and presented as mean ± SEM. A representative kidney PAS-stained image of a specific fungal group as acquired by Leica microscope at ×40 with a 10 µm scale bar was shown. The black arrows indicate the fungal cells. (C) Survival assay of CNA25 (1°—IV) immunized and sham immunized (1°—IV saline) BALB/c mice ($n = 8$) upon an intravenous lethal re-challenge (2°—IV) with *C. albicans* ($5 \times 10^5$ CFU/mice), *C. parapsilosis* ($6 \times 10^6$ CFU/ mice), *C. tropicalis* ($6 \times 10^6$ CFU/mice) and *C. glabrata* ($1 \times 10^7$ CFU/mice) along with saline control was conducted for 30 days and Kaplan–Meier survival graph was plotted. All the fungal cells were live except where HK denotes heat-killed. (D) BALB/c mice ($n = 8$) were immunized with CNA25 ($5 \times 10^5$ CFU/mice) via various routes like oral gavage (1°—Oral), subcutaneous (1°—SC) and intraperitoneal (1°—IP), and after 30 days, they were re-challenged intravenously with WT *C. albicans* ($5 \times 10^5$ CFU/mice, 2°-IV) along with respective saline control groups and monitored for 30 days and Kaplan–Meier survival plot was generated. 1° is primary and 2° denotes re-challenge. Data are representative of two different sets of experiments and are analyzed using the log-rank (Mantel–Cox) test. The actual *P* values are listed in the graph as per the corresponding comparison groups. The numerals in the brackets in each Kaplan–Meier plot represent the number of mice that succumbed on the given day. CFU counts in NAC species infected groups were analyzed in comparison to *C. albicans* infected mice using the Mann–Whitney *U* test (***$P \leq 0.001$).

injected either with saline or $5 \times 10^5$ of WT or CNA25 *C. albicans* cells per mouse and eight mice from each group were euthanized at indicated time points. Similarly, two more vaccinated groups of mice (sham and CNA25 immunization) were generated and after 30 days, they were re-inoculated without (saline) or with WT *C. albicans*. Blood was taken out just before their sacrifice and the tissues like the spleen, lungs, brain, and kidney were excised after sacrificing for subsequent processing (Fig. 2A). First, we determined the fungal load in various organs by CFU analyses and PAS staining of kidneys and found that the unvaccinated mice carried a high fungal load of *C. albicans* (orange line) in the kidney ($\geq 10^5$ CFU per organ), lungs ($4 \times 10^3$ CFU per organ) and brain ($4 \times 10^2$ CFU per organ) within 3 h of inoculation which gradually increased in the kidney ($\sim 3 \times 10^5$ CFU per tissue) but get reduced in other organs (100–200 CFU per organ) in 7 days post infection (Fig. 2Bi–iii). Since no or minimal number of mice infected with WT *C. albicans* survived beyond this period to the next sacrifice period of 15 days, we believed that this fungal load was high enough to cause organ failures leading to death. However, compared to WT, mice infected with CNA25 had considerably reduced fungal load in all of these tissues from 3 h to 7 days of post infections (<4000–200 cells, green line), and all the fungal cells were cleared from organs within 15 days that allowed animals to survive. As expected, the kidney had a maximal fungal load than any other organs even in CNA25-challenged animals. Interestingly, despite the WT re-challenge, the CNA25-vaccinated mice (1°CNA25-2°WT, purple line) possessed a low fungal load in essential organs (<15 × $10^3$ cells per kidney) and almost nil in 30 days post re-challenge, suggesting efficient clearance of pathogenic fungal cells likely due to protective immune responses generated by CNA25. Saline control groups did not show any fungal load at any given time. Regression analyses of fungal load in the kidney of each mouse with respect to days of survival suggested a negative correlation between fungal load and survivability (Fig. EV1B). CFU analyses of kidneys of various groups of animals were further authenticated by PAS staining, and only the primary WT-challenged group mice kidneys showed the presence of hyphal *C. albicans* cells, and due to a low or negligible fungal load in other groups of mice, *C. albicans* cells were not detected (Fig. EV1C).

## Altered systemic levels of immune cells upon *C. albicans* infections

Abnormality in blood cell count indicates infection. On the other hand, immune suppression due to a quantitative and qualitative change in immune cells is associated with the development of candidiasis (Gow et al, 2012). To get an overview of systemic circulation levels of various immune cells upon *C. albicans* infection, blood from different groups of mice was analyzed using a blood cell analyzer, and various immune cell counts were determined (Fig. 3A). The time kinetics of most of the immune cells followed distinct trends in virulent WT and avirulent CNA25-infected mice (compare orange and green lines). We did not find any change in total WBC, monocyte, and granulocyte counts after 3 h post infection in any of the groups; however, they increased after day 3 in WT and CNA25-infected mice. While these cell counts peaked on day 7 only in the WT-infected mice that faced 100% mortality within 10 days of post infection, interestingly, in CNA25-infected mice these cell counts started declining by day 7

towards basal levels until day 30 which is associated with 100% survival of the mice. Analysis of lymphocyte count showed an increase to a peak as early as 3 h that maintained until day 3 and declined to a normal level by day 7 until later duration of post infection with CNA25. Whereas in WT-infected mice, the lymphocyte count marginally reduced than the basal level in the time kinetics suggesting the onset of immunosuppression. Notably, the platelet counts increased at 3 h of post infection in the fungal-challenged groups, although they further increased on the 3rd day and subsided gradually to basal level in the CNA25-challenged group but not in WT-infected mice. Even in the CNA25-immunized mice, when they were re-challenged with WT *C. albicans* (1°CNA25-2°WT), most of these blood cell counts increased till 15 days but not as high as in the WT primary-challenged mice, and then they gradually reduced to basal counts (purple line). The simple linear regression analyses of blood cells in each infected mouse vs. days of infection suggested two distinct trends in WT and CNA25 mice groups. While WBC, monocytes, granulocytes, and platelets showed a steady upward trend with respect to days of infection with WT *C. albicans*, a gradual downward trend of these cells was observed in CNA25 primary-challenged and vaccinated mice groups (Fig. 3B). Interestingly, disease severity was correlated with a drop in lymphocyte counts in WT-challenged mice. Taken altogether, it is evident that CNA25 induced a certain level of early inflammation without enhancing disease progression, but that subsided later, a typical characteristic of a vaccine strain; whereas both hyper inflammation and immunosuppression seem to be associated with disease severity of WT *C. albicans* infection.

## Efficient immune clearance of CNA25 by macrophages

Innate immune cells such as macrophages and neutrophils are the first line of defense system against invading pathogens. Macrophages first engulf *C. albicans* cells and deliver them to the phagolysosome for clearance. However, *C. albicans* can evade macrophage clearance by developing hyphae and releasing various cytolytic toxins and enzymes (da Silva Dantas et al, 2016). Next, we examined the interaction of fungal and macrophage cells in a co-culture experiment where we measured the efficiency of phagocytosis, fungal clearance by macrophages, and immune evasion of fungi (Fig. 4). The deep-red-stained RAW 264.7 murine macrophage cells and CFSE-stained *C. albicans* cells were mixed in a 1:1 ratio and co-cultured, and cells were taken up for flow cytometry at the mentioned time points. Using a proper gating strategy (Appendix Fig. S1A), fungal-associated macrophage cells (double-stain-positive, pink color) were estimated (Fig. 4A). Consistent with the reported results, we observed ~90% of macrophages uptaking WT fungal cells within 1 h of co-culturing (Kumari et al, 2023). The efficiency of phagocytosis was reduced to 78–73% upon longer incubation. However, relatively low phagocytosis was observed when CNA25 was mixed with macrophages at each time point (80–50%). The reduced double-stain-positive cells in CNA25-macrophage co-culture could be due to a faster clearance of fungal cells and vice versa in WT-macrophage co-culture. *C. albicans* colony-forming units (CFU) ability of the co-cultures was verified and the result revealed a faster clearance of CNA25 than the WT cells (Fig. 4B). Conversely, we observed a higher population of macrophage cell death when RAW cells were co-cultured with WT than with CNA25 fungal cells (Fig. 4Ci,ii). As PI stain only enters the cells with a leaky

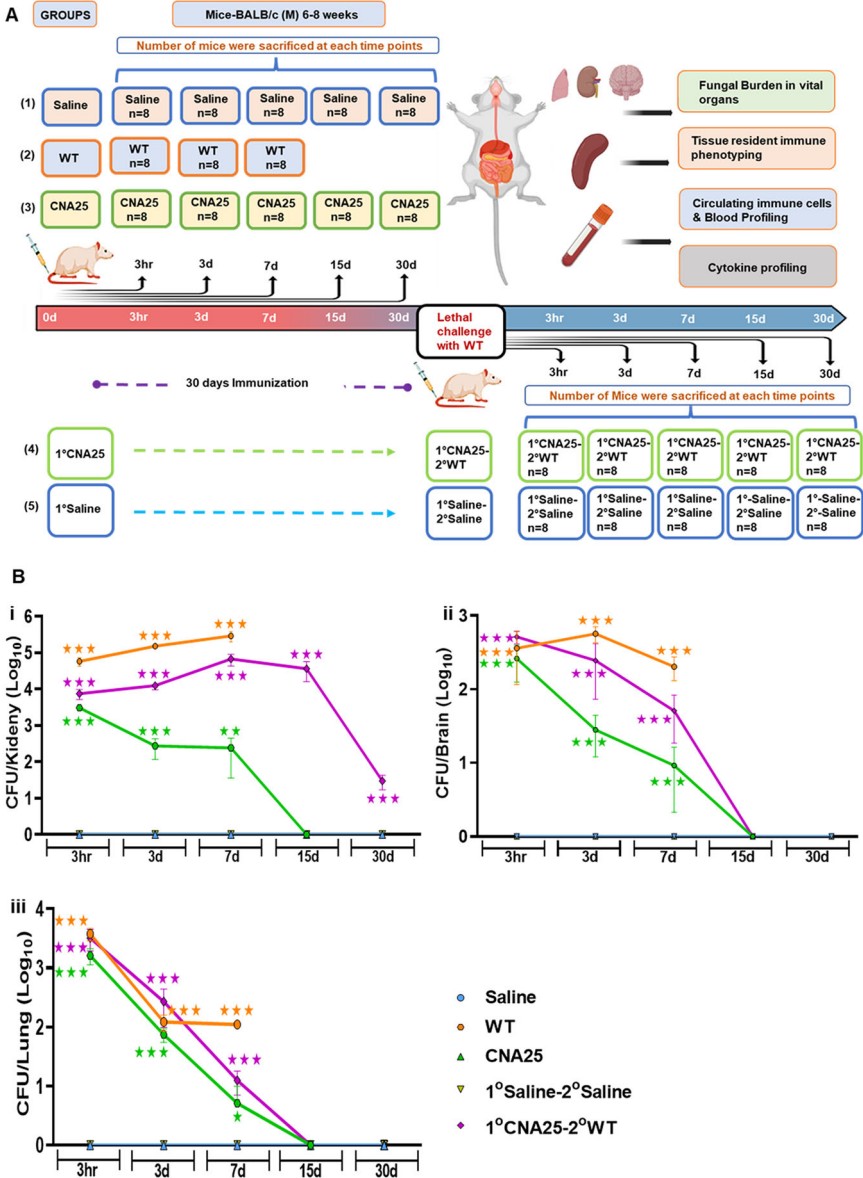

**Figure 2. Kinetic analyses to determine fungal burden in various mice groups.**

(A) A schematic representation of complete kinetic experiment for 30 days except for the WT primary-challenged mice group in pre- and post-immunization set up. For the WT-inoculated mice group, up to 7 days of kinetic analysis was carried out. BALB/c mice were intravenously injected with WT ($5 \times 10^5$ CFU/mice), CNA25 ($5 \times 10^5$ CFU/mice), and saline (100 µl $1 \times$ PBS) on the 0th day and pre-immunization kinetic studies were performed (upper section). For the post-immunization study, vaccinated BALB/c mice groups were generated by IV administration with CNA25 ($5 \times 10^5$ CFU/mice) or saline control (100 µl $1 \times$ PBS) on the 0th day. Post 30 days, mice were re-challenged with $5 \times 10^5$ CFU/mice of WT or saline (lower section). Mice were (8 mice /group) euthanized at indicated time points (3h, 3d, 7d, 15d, and 30d) in both pre- and post-immunization groups, and the blood, spleen, lungs, and kidney were collected. (B) Fungal burden was accessed by CFU analysis of the kidneys (i), brain (ii), and lungs (iii) organs. A multicolored line graph of the pooled data with mean ± SEM obtained for each mice group (WT—Orange, CNA25—green, saline—sky blue, 1°saline-2°saline—greenish yellow, and 1°CNA25-2°WT—purple) at each time of sacrifice is shown. Data are representative of two separate experiments and were analyzed in comparison to saline using the Mann–Whitney $U$ test. *$P \leq 0.05$, **$P \leq 0.01$, and ***$P \leq 0.001$ are also colored similar to the mice group.

membrane, a characteristic of dead cells, more PI-positive cells were observed under the fluorescence microscope for RAW-WT co-culture, indicating a higher percentage of macrophage death. Altogether, our results implemented that CNA25 failed to evade the immune system effectively, whereas a higher percentage of wild-type *C. albicans* cells evaded the immune system and escaped by inducing macrophage killing.

## Differential expression profile of pattern recognition receptors (PRRs) in fungal-infected mice

Fungal cells are first recognized by pattern recognition receptors (PRRs) of myeloid cells. The cell wall, which is composed of inner chitin, middle β-glucans, and upper mannan layers, contributes significantly to fungal recognition and transduction of downstream

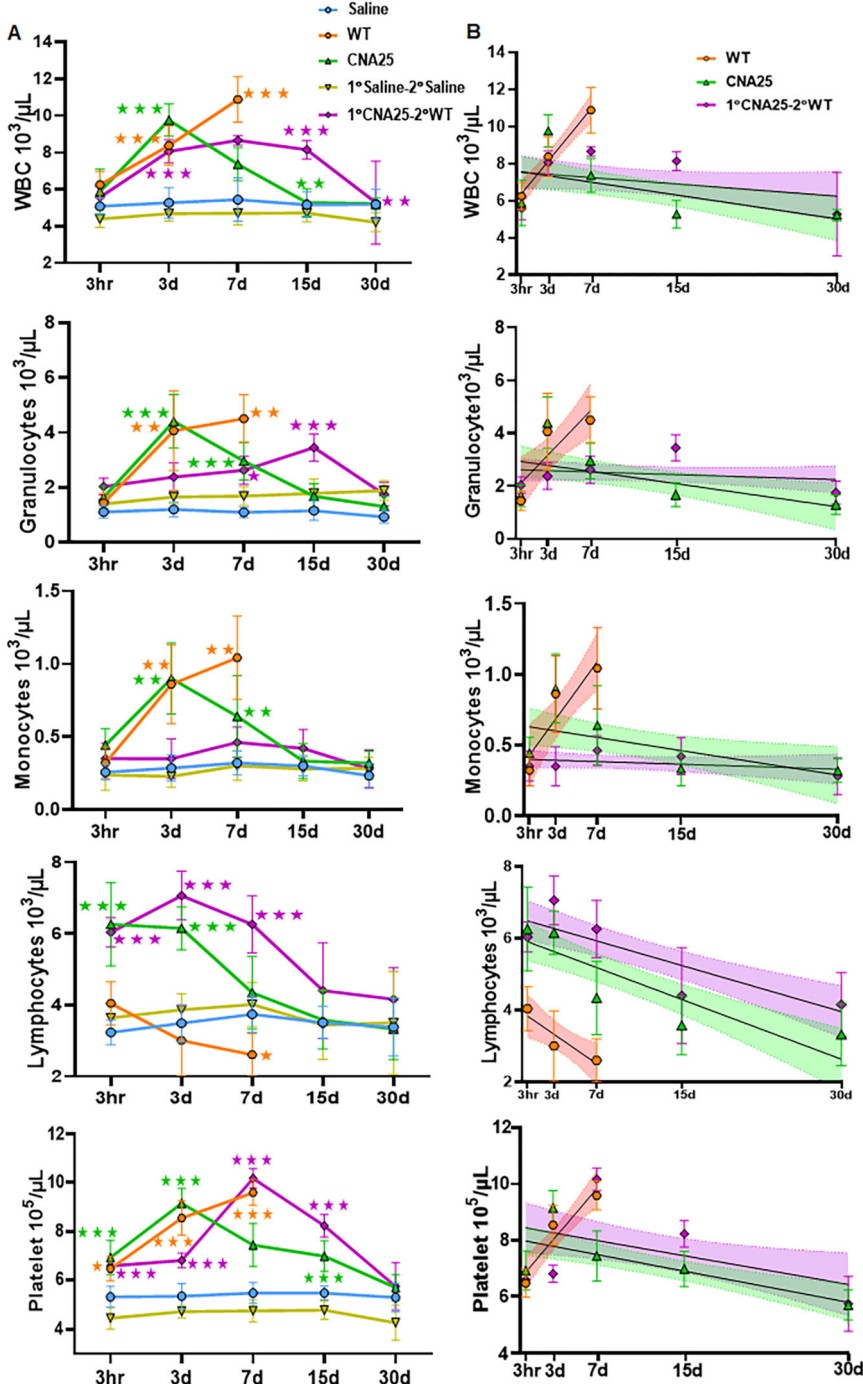

**Figure 3. Complete blood profiling.**

(A) Blood cells like WBC, granulocytes, monocytes, lymphocytes, and platelets of each group of mice (8 mice/per group) as indicated in Fig. 2A were counted by a blood analyzer. The multicolored line graph connecting the average cell counts of replicates in each mice group (WT—Orange, CNA25—green, saline—sky blue, 1°saline-2°saline —greenish yellow, and 1°CNA25-2°WT—purple) at each time of sacrifice is shown. Data are the representative of two separate experiments, presented as mean ± SEM, and the statistical comparison with respect to saline (*$P \leq 0.05$, **$P \leq 0.01$, ***$P \leq 0.001$, and no star mark-nonsignificant) as determined using the two-way ANOVA (Tukey's multiple comparisons test) is shown. (B) A linear regression and Person's correlation analyses of various immune cell counts versus days post-fungal infection in individual mice of WT (orange), CNA25 (green), and 1°CNA25-2°WT (purple) groups are provided and parameters obtained from the analyses are given in Table EV2. Data were from eight biological replicates and presented as mean ± SEM.

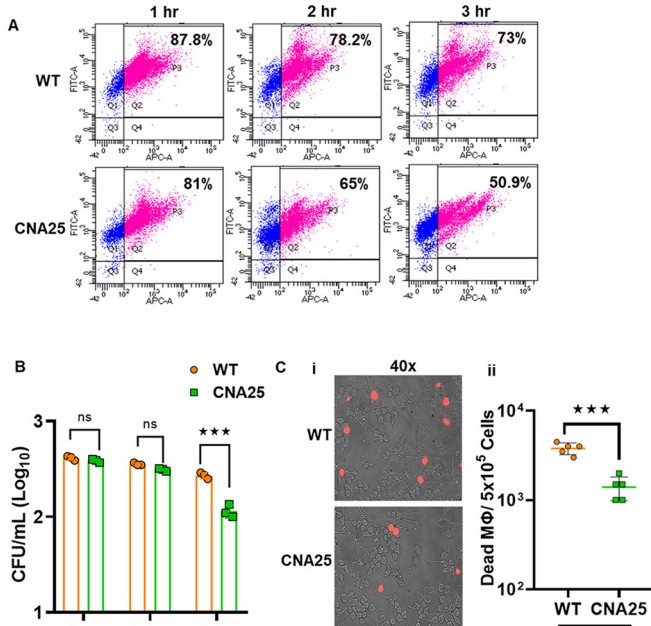

**Figure 4. Macrophage-fungal cell interaction.**

(A) CFSE-labeled WT and CNA25 were co-cultured with deep-red-stained murine macrophage cells in a 1:1 ratio for 3 h. At regular intervals, cells were pooled, acquired, and double-positive cells (CFSE-FITC channel and Deep Red-APC channel; Q2, P3) were analyzed in BD LSRFortessa and representative bi-variant plots are shown. (B) After each time point, WT and CNA25 cells were extracted out of the macrophages from the co-cultures, and fugal cell death was determined by estimating CFUs. Data were from three repeats and presented as mean ± SEM. (C) Macrophage killing induced by fungal cells was determined by staining the co-cultured cells of 2.5 h incubation with PI. Representative merged PI-stained microscopic images with 75 μm scale bar (EVOS imaging system, × 40) of co-cultured cells are shown (i). The graph with mean ± SEM of PI-stained macrophages with five biological replicates of WT and CNA25 independent co-cultures is shown (ii). Statistical tests two-way ANOVA and Chi-square (and Fisher's extract) test were performed to compare the CFU and macrophage dead population, respectively (***$P \leq 0.001$).

signaling to activate the host immune system (d'Enfert et al, 2021; McKenzie et al, 2010). While Dectin-1/Clec7a binds β-glucans, Dectin-2, TLRs (TLR2/4), and mannose receptors recognize α-mannans. Similarly, TLR9 and NOD2 receptors recognize chitins. Since our earlier report suggested differential cell wall compositions in WT and CNA25 cells (Patel et al, 2023), we checked the expression of some of these PRRs after isolating RNA from the infected kidneys by qRT-PCR (Fig. 5A). Kidney was selected for this analysis as it carries maximal fungal load and high immune cells infiltration. In addition to infiltrated immune cells, renal epithelial cells also express some of these PRRs (Cohen-Kedar et al, 2014; Decanis et al, 2009; Swidergall et al, 2018; Weindl et al, 2007). Renal transcription analysis revealed that the expression of Dectin-1, TLR2, TLR4, and MYD88 increased within 3 h of post-inoculation and gradually decreased at later time points in the WT primary-challenged mice. While the fungal load was high on day 3 and day 7, a lower expression of these PRRs during these periods in WT-infected mice was observed, again indicating active immune evasion by virulent fungal cells. In contrast, the mRNA expression of these receptors showed a delayed increase in the CNA25-infected group and reached to maximum at later time points

(days 7 and 15). It suggested that although immune recognition and possibly downstream activation and signaling take place very early in the WT-challenged mice, and later on they get suppressed, in the vaccine strain challenged mice those events started late but remained steady. TLR2 receptor functions through MYD88 cytosolic factor that activates MAPK/AP1 cascade to produce various pro-inflammatory cytokines. Our transcriptional analysis suggested that MYD88 level also increased similar to TLR2 as in WT and CNA25-infected mice. Since Dectin-1 and TLR2 remain activated for a longer period than other PRRs, they may play a critical role in steady immune system activation in the CNA25-infected group. To further demonstrate the role of these receptors in protective immunity, we decided to block Dectin-1 and TLR2 receptors by injecting anti-Dectin-1 and anti-TLR2 antibodies to the CNA25-vaccinated mice ($n = 6$) at regular intervals and then re-infected with WT *C. albicans* and survivability was monitored (Fig. 5Bi,ii). The efficiency of blocking of these receptors (~70–90%) in WBCs isolated from the blood of the CNA25-immunized mice was confirmed by flow cytometry (Appendix Fig. S2A,B). While ~85% of the immunized mice blocked with anti-Dectin-1 antibody succumbed to reinfection ($P = 0.0005$) with a median death time of 9 days, anti-TLR2 antibody-mediated blocking had minimal effect on resistance to lethal challenge as 85% of mice were protected. Isotype antibody-mediated control blocking did not alter the survival of immunized mice to lethal re-challenge. Fungal sepsis in the killed mice was confirmed by CFU score (Appendix Fig. S4A). This result suggested that Dectin-1 plays a critical role in fungal cell recognition and long-term immune cell activation in the CNA25-vaccinated mice, while other PRRs like TLR2 may assist in recognition in the early stage of infection.

## Enhanced neutrophils, natural killer, B, and T cells are critical for antifungal activity in infected mice

Circulating and splenic immune cell dynamics reflect the strength of pathogen-immune system interaction and disease progression. To understand the dynamics, WBC from blood and splenocytes from the spleen of the fungal-infected mice were isolated and immune-phenotyping was carried out using a flow cytometer (Fig. 6A,B). Immune cells were stained with specific fluorescent-tagged antibodies and using a suitable gating strategy cells were segregated (Appendix Fig. S1B–E; Figs. EV2 and EV3). Major immune cell alteration started within 3 h of infection and lasted up to 15 days, and on the 30th day, a basal level of cells was observed in most of the analyzed mice. Since the mice infected with WT *C. albicans* die between 7 and 15 days, no data was collected for the 15th and 30th day periods. Circulating myeloid macrophages + DCs (CD11b⁺/Gr-1⁻/SSC(lo)) population did not alter significantly in the WT-infected mice as opposed to saline control mice, however, they increased in the CNA25 inoculated group and remained high in the 3rd day. At later time points (7-30 days), their counts did not differ among the mice groups (Fig. 6Ai). Interestingly, the splenic macrophages + DCs mostly increased in the mice infected with any fungal strain irrespective of their virulence status at all time points except on the 30th day (Fig. 6Bi). On the other hand, both circulating and splenic neutrophils (CD11b⁺/Gr-1⁺/ʰⁱ/SSC(hi)) popu-lations only altered and increased mostly in the mice group vaccinated with CNA25 and re-challenged with WT (1°CNA25-2°WT, purple line; Fig. 6Aii,Bii). While the circulating B cells (CD11b⁻/CD19⁺/CD5⁺/⁻) increased in mice infected with any fungal strains, peripheral T cells (CD11b⁻/CD19⁻/CD5⁺) population reduced in mice only when

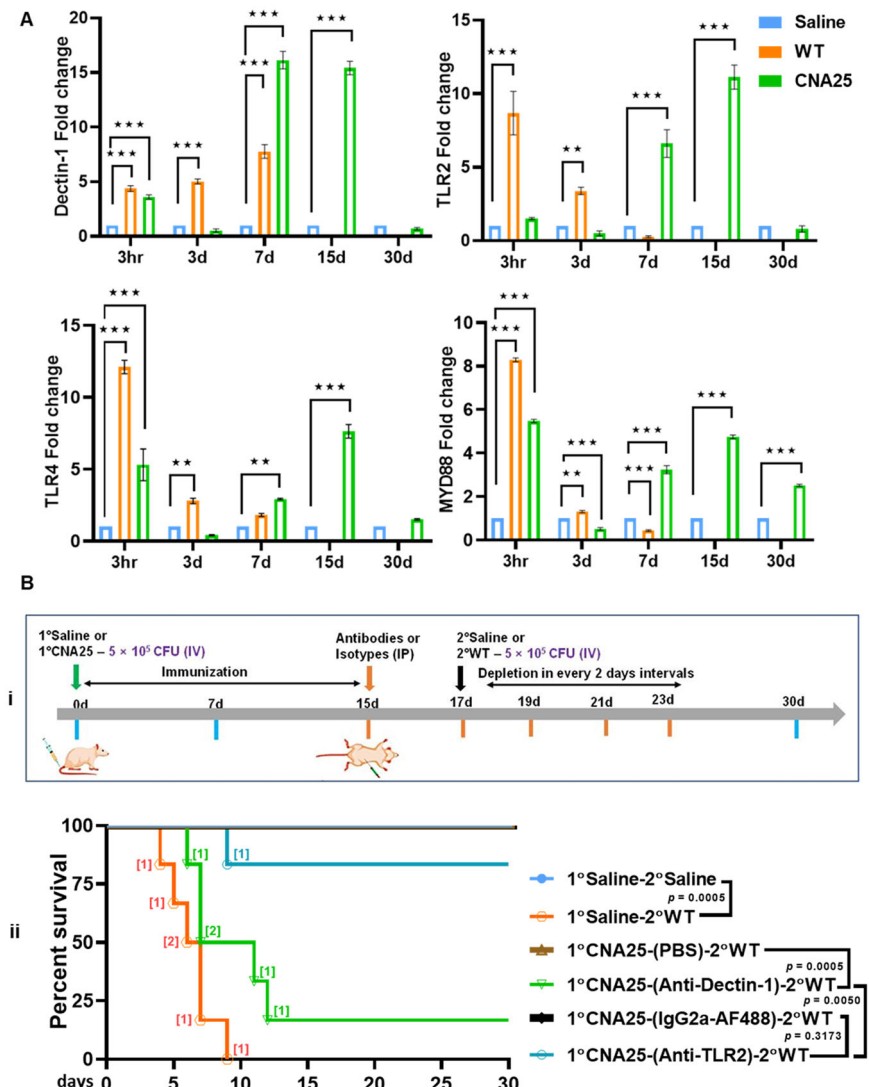

**Figure 5. Gene expression analysis of PRRs in renal tissue by qRT-PCR and their in vivo role.**

(A) Expression of PRRs such as Dectin-1, TLR2, TLR4, and MYD88 in the infected kidneys of pre-immunized mice groups (WT, CNA25, and Saline) was determined by qRT-PCR. Expression of the genes of interest was normalized with *HPRT1* and the fold change was estimated by calculating $2^{-\Delta\Delta Ct}$. The mean ± SEM of eight mice was shown per group in the bar graph. Data are the commulative of two separate experiments. Two-way ANOVA was used to compare the fold change (**$P \leq 0.01$, ***$P \leq 0.001$, no star marks nonsignificant). (B) (i) A schematic representation of in vivo blocking assay of PRRs by specific monoclonal antibodies and respective isotypes. After 15 days of immunization, the mice ($n = 6$/group) were intraperitoneally (IP) injected with the required antibody or isotype. Two days after 1st round of depletion, mice were re-challenged with WT or saline. After 2 h, 2nd round of antibody administration was carried out, which was followed by three successive depletions at every two-day interval. Survival was monitored for all mice up to 30 days. (ii) A Kaplan–Meier survival curve was generated to compare the survival rates of Dectin-1 depleted (1°CNA25-(Anti-Dectin-1)-2°WT) and TLR2-depleted (1°CNA25-(Anti-TLR2)-2°WT) groups of mice with their respective PBS and isotype control counterparts (1°CNA25-(PBS)-2°WT and 1°CNA25-(IgG2a-AF488)-2°WT). This experiment was also conducted in the presence of saline (1°Saline-2°Saline) and WT lethal control (1°Saline-2°WT). Statistical significance was determined using the log-rank (Mantel–Cox) test. The P values are listed beside the corresponding comparison groups.

the WT *C. albicans* was inoculated, otherwise, their level increased in CNA25-challenged groups (Fig. 6Aiii,iv). The resident B-cell counts mostly altered in the WT re-challenged CNA25-immunized mice group, whereas the splenic T cells again gradually reduced after an increase at 3 h post infection in the WT primary-challenged group. The T-cell population increased steadily upon avirulent strain challenge (Fig. 6Biii,iv). Similarly, spleen-specific CD5$^+$ B-cell populations (also known as B1 cells) were induced mostly in the vaccinated mice but reduced upon WT re-challenge (Fig. 6Bv). Natural

killer (NK) cells (CD43$^-$/CD161$^+$) also contribute to the rapid innate immune response against invading pathogens (Schmidt et al, 2017) and both tissue-resident NK and CD43$^+$ NK cells (CD43$^+$/CD161$^+$) distribution elevated in vaccine strain inoculated mice (Fig. 6Bvi,vii; Appendix Fig. S3A). Together, these results suggested that WT *C. albicans* induces immune suppression by lowering T-cell count, whereas an increased population of neutrophils, NK, B, and T cells in CNA25 strain infected mice are most likely involved in fungal clearance and protective immunity.

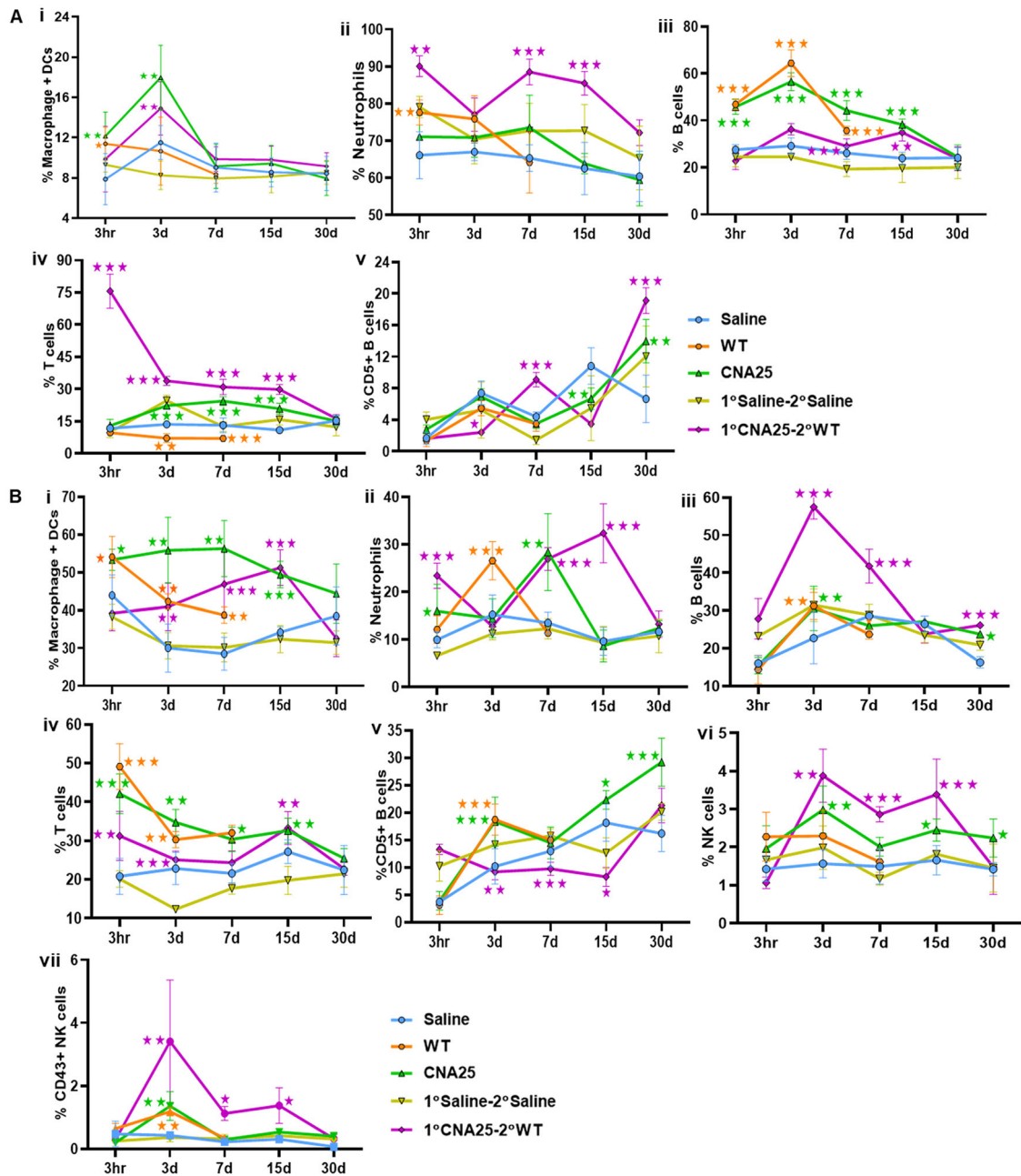

**Figure 6. Phenotyping of myeloid and lymphoid cell populations in various mice groups.**

(A) The multicolored line graphs showing the kinetic alteration of circulating macrophage + DCs (i), neutrophils (ii), B cells (iii), T cells (iv), and CD5+ B cells (v) population in mice groups (WT—orange, CNA25—green, saline—sky blue, 1°saline-2°saline—greenish yellow and 1°CNA25-2°WT—purple) at various time points are plotted. The lines join the mean± SEM of eight mice data for each group and time point. Data are representative of two separate experiments and were analyzed using the two-way ANOVA test. *P ≤ 0.05, **P ≤ 0.01, ***P ≤ 0.001. (B) The multicolored line graphs showing the kinetic alteration of splenic-resident macrophage + DCs (i), neutrophils (ii), B cells (iii), T cells (iv), CD5+ B cells (v), NK cells (vi), and CD43+ NK cells (vii) population in mice groups (WT—Orange, CNA25—green, saline—sky blue, 1°saline-2°saline—greenish yellow and 1°CNA25-2°WT—purple) at various time points are plotted. The lines join the mean± SEM of eight mice data for each group and time point. Data are representative of two separate experiments and were analyzed using the two-way ANOVA test (Tukey's multiple comparisons test). *P ≤ 0.05, **P ≤ 0.01, ***P ≤ 0.001.

## INFγ, TNFα, and IL-17-positive CD4+ T cells are critical for immunity against *C. albicans* infection

CD4 and CD8 lineages are the αβ T cells involved in T cells mediated adaptive immune responses. Our flow cytometry analysis revealed a higher population of both CD4+ and CD8+ T cells in WT and CNA25 primary-challenged mice at 3 h post infection, and subsequently, they reduced (Fig. 7Ai,ii; Appendix Fig. S3B). However, an elevated population of these T cells remained throughout the duration in the WT re-challenged CNA25-

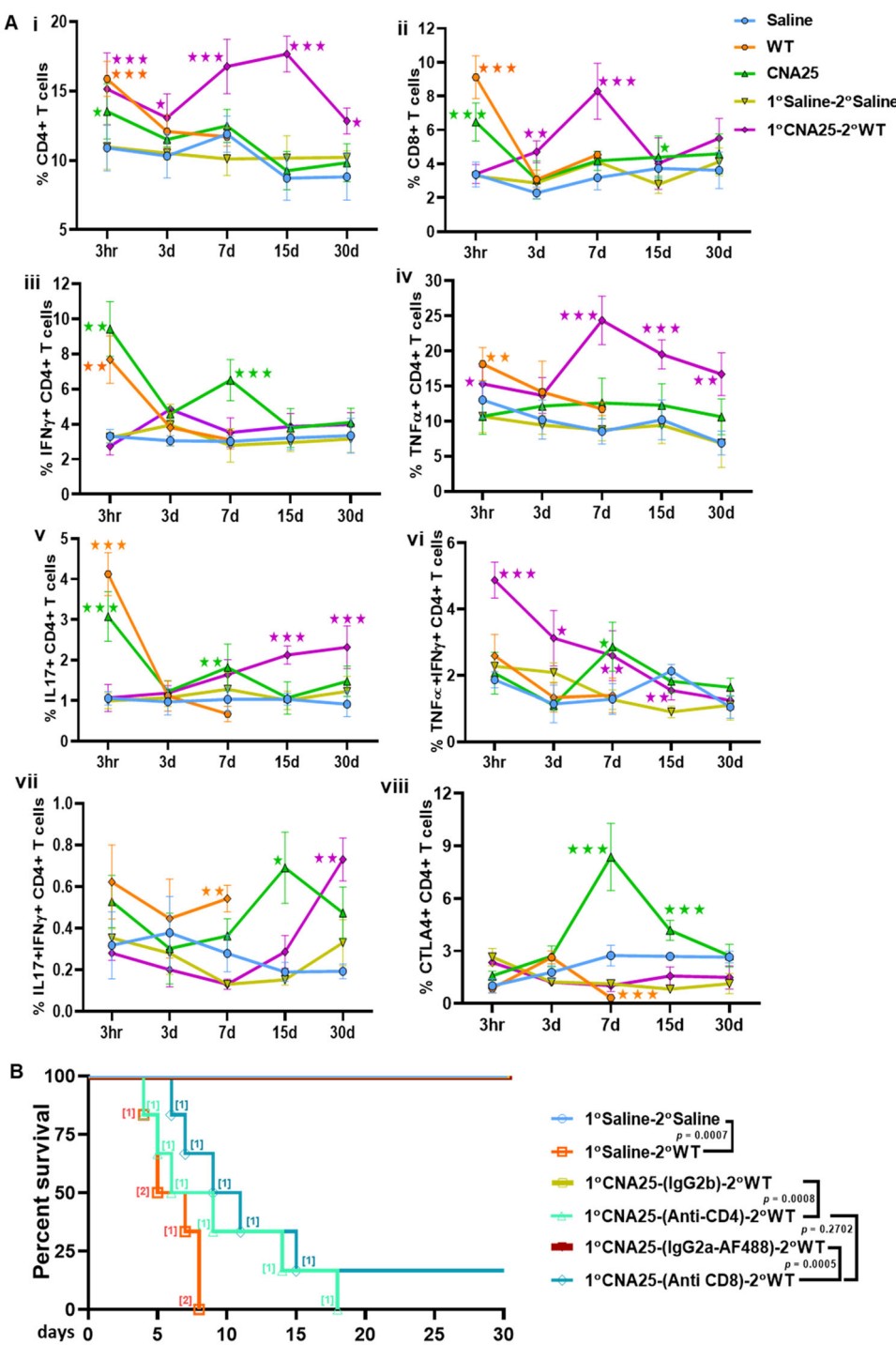

**Figure 7. T-cell subsets determination in splenocytes.**

(A) The multicolored line graphs showing the kinetic alteration of splenic-resident CD4$^+$ T cells (i), CD8$^+$ T cells (ii), IFNγ$^+$ CD4$^+$ T cells (iii), TNFα$^+$ CD4$^+$ T cells (iv), IL-17$^+$ CD4$^+$ T cells (v), TNFα$^+$ IFNγ$^+$ CD4$^+$ T cells (vi), IL-17$^+$IFNγ$^+$ CD4$^+$ T cells (vii), and CTLA4$^+$ CD4$^+$ T cells (viii) population in mice groups (WT—orange, CNA25—green, saline—sky blue, 1°saline-2°saline—greenish yellow and 1°CNA25-2°WT—purple) at various time points are plotted. The lines join the mean± SEM of eight mice data for each group and time point. Data are representative of two separate experiments and were analyzed using the two-way ANOVA test (Tukey's multiple comparisons test). *$P \leq 0.05$, **$P \leq 0.01$, ***$P \leq 0.001$. (B) A Kaplan–Meier survival curve was generated to compare the survival rates of CD4-depleted (1°CNA25-(Anti-CD4)-2°WT) and CD8-depleted (1°CNA25-(Anti-CD8)-2°WT) groups of mice ($n = 6$/group) with their respective isotype control counterparts (1°CNA25-(IgG2b)-2°WT and 1°CNA25-(IgG2a-AF488)-2°WT). This experiment was also conducted in the presence of saline (1°Saline-2°Saline) and WT lethal control (1°Saline-2°WT). Statistical significance was determined using the log-rank (Mantel–Cox) test. The P values are listed beside the corresponding comparison groups.

immunized mice group (1°CNA25-2°WT, purple line), albeit a relatively smaller population of CD8$^+$ T cells. The precise function of CD8$^+$ T cells in antifungal immunity is yet to be established; however, several reports suggest its role in fungal growth inhibition by producing IFNγ (Cui et al, 2013; Lindell et al, 2005). To determine the role of CD4$^+$ and CD8$^+$ T cells mediated adaptive immunity in antifungal resistance, a cellular depletion assay was performed where we injected anti-CD4 and anti-CD8 antibodies along with respective isotype antibody controls to the CNA25-immunized mice and susceptibility to lethal *C. albicans* challenge was monitored as depicted in Fig. 5Bi. The cellular depletion of these cells in splenocytes was confirmed by flow cytometry (Appendix Fig. S2C,D). While the control antibodies given mice were resistant to a lethal dose of WT *C. albicans*, upon cellular ablation of CD4$^+$ and CD8$^+$ T cells, the CNA25-vaccinated mice succumbed to re-challenge ($P = 0.0005$–0.0008) ascertaining the critical role of both cells in protective fungal immunity (Fig. 7B). In comparison to CD4$^+$ depleted mice (0%, MDT = 7 days), CD8$^+$ depleted immunized mice showed better protection (20%, MDT = 10 days). A high fungal load in the kidneys of these killed mice confirmed death due to fungal sepsis (Appendix Fig. S4B). Further analysis revealed that TNFα$^+$, INFγ$^+$TNFα$^+$ double-positive, and IL-17$^+$ CD4$^+$ T cells population remained high in CNA25-vaccinated re-infected with WT group (Figs. 7Aiv,v,vi and EV4i,ii). INFγ, TNFα, and IL-17-positive CD4$^+$ T cells population only increased at an early stage of infection by WT and CNA25 suggesting their role in fungal clearance and early immune activation (Fig. 7Aiii,iv,v). Interestingly, a delayed increase of IL-17$^+$INFγ$^+$ double-positive T cells population was observed mostly in CNA25-vaccinated mice groups (Fig. 7Avii). The response of T cells is controlled by multiple checkpoints to avoid unwanted activation and CTLA4 is one such protein that inhibits re-activation of T cells (Lee et al, 1998). We found that CTLA4$^+$ T cells were elevated only in the mice group infected with CNA25 after 7 days of post infection and later on it reduced to basal level. In other groups of mice, it did not alter significantly (Fig. 7viii; Appendix Fig. S3C). This result again confirmed differential virulence attributes of CNA25 and WT *C. albicans* strains and the involvement of CD4$^+$ and CD8$^+$ T cells mediated adaptive immune responses to fungal infection.

## Differential induction of systemic cytokines upon fungal infections and their role in protective immunity

Cytokines are messengers of the immune system, promptly and transiently produced by the immune cells in response to infections, and play critical roles in regulating host responses to infectious agents and inflammatory stimuli. Since we observed differential counts of various circulating and splenic immune cells in various groups of mice, the levels of serum cytokine and chemokine might also differ. Using a Bio-plex analyzer, we estimated 21 different cytokines (Figs. 8 and EV5). We observed induction of all the cytokine and chemokine levels in the fungal-infected mice irrespective of their virulence status, although with different time kinetics. The levels of pro-inflammatory cytokines such as IL-1α, IL-1β, IL-2, IL-6, IL-12, IL-17, INFγ, and TNFα increased in the serum of mice infected with fungal strains from a very early stage (3 h) to 7 days post infection (Fig. 8A). The WT-infected mice produced a very high level of these pro-inflammatory cytokines and

their maximum levels reached in 7 days post-challenge which is correlated with high fungal load in comparison to in vaccinated groups. A high inflammatory state due to the induction of pro-inflammatory cytokines could induce tissue damage and is responsible for death in WT primary-challenged mice (Orange Line). After the 7th day, these cytokine levels gradually reduced in CNA25 primary-challenged mice to the basal level to maintain immune homeostasis, although slightly higher levels in the WT re-challenged vaccinated group (1°CNA25-2°WT). Anti-inflammatory cytokines such as IL-3, IL-4, IL-5, IL-9, IL-10, and IL-13 were also induced within 3 h of fungal infection in all three groups of mice and remained high up to the 7th day period (Fig. 8B). In the WT re-challenged mice group, these cytokines except IL-9 were significantly high till 30 days post infection and they could be responsible for protective immunity. Maximum induction of IL-9 was observed in CNA25 primary-challenged group mice through-out the study duration. IL-4 was only induced in secondary WT re-infected immunized mice again highlighting its involvement in protective immunity. A very high level of pro-inflammatory cytokines such as IL-10 and IL-13 at 7 days WT-infected mice group suggested a high degree of immunosuppression. The levels of chemokines like RANTES, G-CSF, MCP-1, MIP-1 (α and β), and KC were found to be high in the serum of fungal-infected mice from 3 h to 7 days post infection periods and gradually they suppressed except in the WT re-challenged group where the level was still significantly high till 30th day (Fig. EV5). To understand the significance of these cytokines, molecular depletion of representative pro- and anti-inflammatory cytokines like IFNγ, TNFα, and IL-17 was carried in CNA25-immunized mice as shown in Fig. 5Bi and survivability was monitored upon WT re-challenged (Fig. 8C). While all the immunized mice with a reduced level of IFNγ and IL-17 succumbed to reinfection ($P = 0.0005$), about 90% mice dead by TNFα depletion. However, the median death time was the same 7 days for the depleted mice. Isotype antibodies did not affect the survival of immunized mice to lethal re-challenge. The severity of fungal sepsis in the killed mice was confirmed by determining the CFU score (Appendix Fig. S4C). Depletion of these cytokines in CD4$^+$ T cells of CNA25-vaccinated mice was confirmed by flow cytometry (Appendix Fig. S2E–G). Taken all together, our data suggests that while a critical balance of the pro- and anti-inflammatory cytokines is maintained for long durations in mice challenged with CNA25 cells resulting in antifungal protective immunity, both of these cytokines remained high in the WT-challenged mice group leading to immunosuppression and inflammatory state allowing disease progression and death.

## Safety and efficacy of CNA25 immunization in immunosuppressed and immunodeficient mice

Next, to check whether CNA25 is also avirulent and can generate protective immune responses even in immunocompromised contexts, we used two different mice models. Mouse strains with severe combined immunodeficiency (SCID) have markedly reduced mature B and T cells as a result of defects in the rearrangement of immunoglobulin and T-cell antigen receptor genes (Bosma et al, 1983). BALB/c SCID mice ($n = 6$) were subjected to intravenous inoculation of *C. albicans* ($5 \times 10^5$ CFU), and their survivability was monitored (Fig. 9Ai). Importantly, even in SCID mice, CNA25 was found to be avirulent as all of the mice survived, thus the other

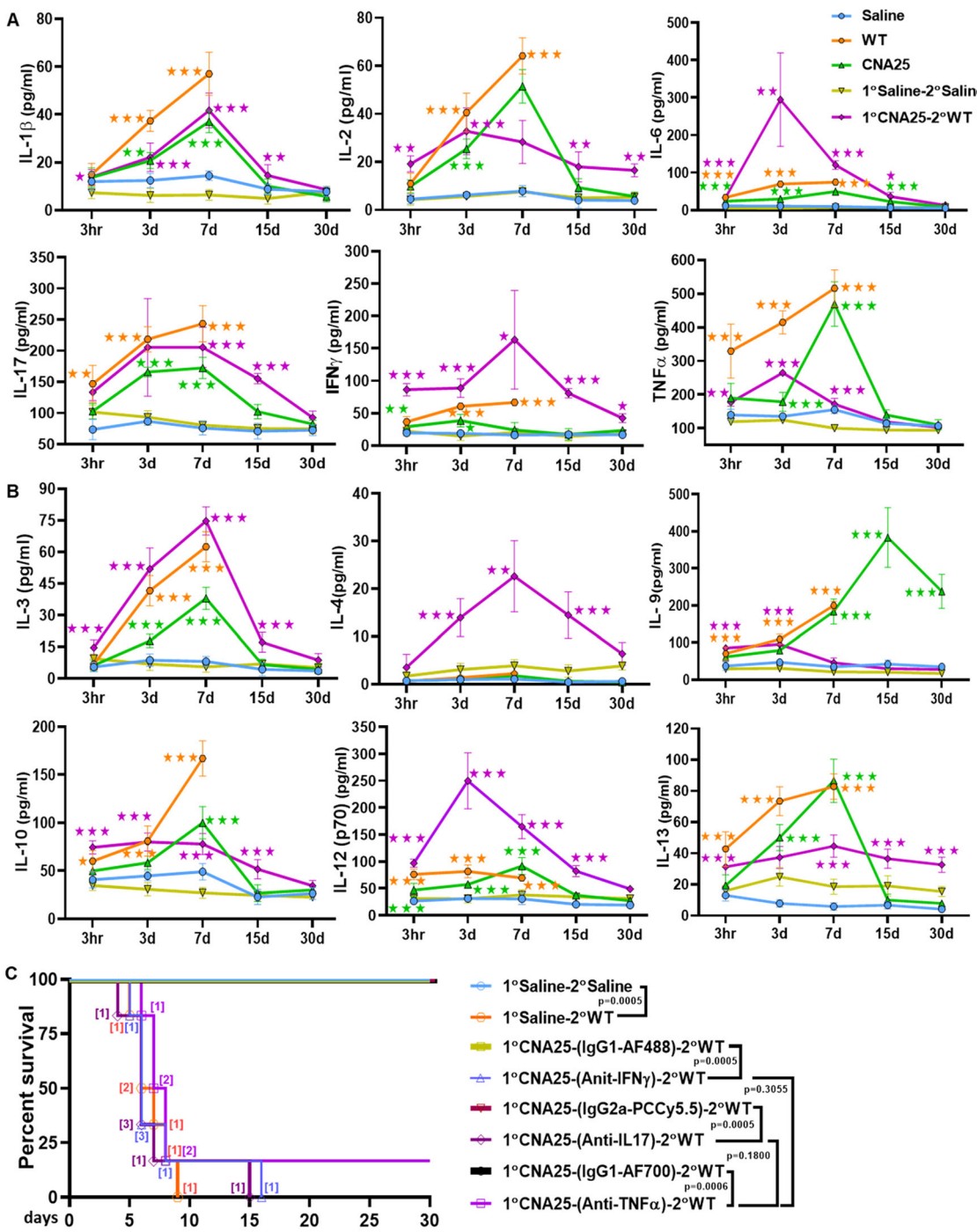

**Figure 8. Serum cytokine profile of infected mice.**

The multicolored line graphs showing the kinetic alteration of (A) pro-inflammatory cytokines (IL-1β, IL-2, IL-6, IL-17, IFNγ, and TNFα), and (B) anti-inflammatory (IL-3, IL-4, IL-9, IL-10, IL-12, and IL-13) levels in picogram/ml in the serum of mouse groups (WT—Orange, CNA25—green, saline—sky blue, 1°saline-2°saline—greenish yellow, and 1°CNA25-2°WT—purple) at defined time points. The lines join the mean± SEM of eight mice data for each group and time point. Data are representative of two separate experiments and were analyzed using the two-way ANOVA test (Tukey's multiple comparisons test). *$P \leq 0.05$, **$P \leq 0.01$, ***$P \leq 0.001$. (C) A Kaplan–Meier survival curve was generated to compare the survival rates of IFNγ-depleted (1°CNA25-(Anit-IFNγ)-2°WT), TNFα-depleted (1°CNA25-(Anti-TNFα)-2°WT) and IL-17-depleted (1°CNA25-(Anti-IL-17)-2°WT) group of mice ($n = 6$/group) with their respective isotype control counterparts (1°CNA25-(IgG1-AF488)-2°WT, 1°CNA25-(IgG1-AF700)-2°WT, and 1°CNA25-(IgG2a-PCCy5.5)-2°WT). This experiment was conducted in the presence of saline (1°Saline-2°Saline) and WT lethal control (1°Saline-2°WT). Statistical significance was determined using the log-rank (Mantel–Cox) test. The P values are listed beside the corresponding comparison groups.

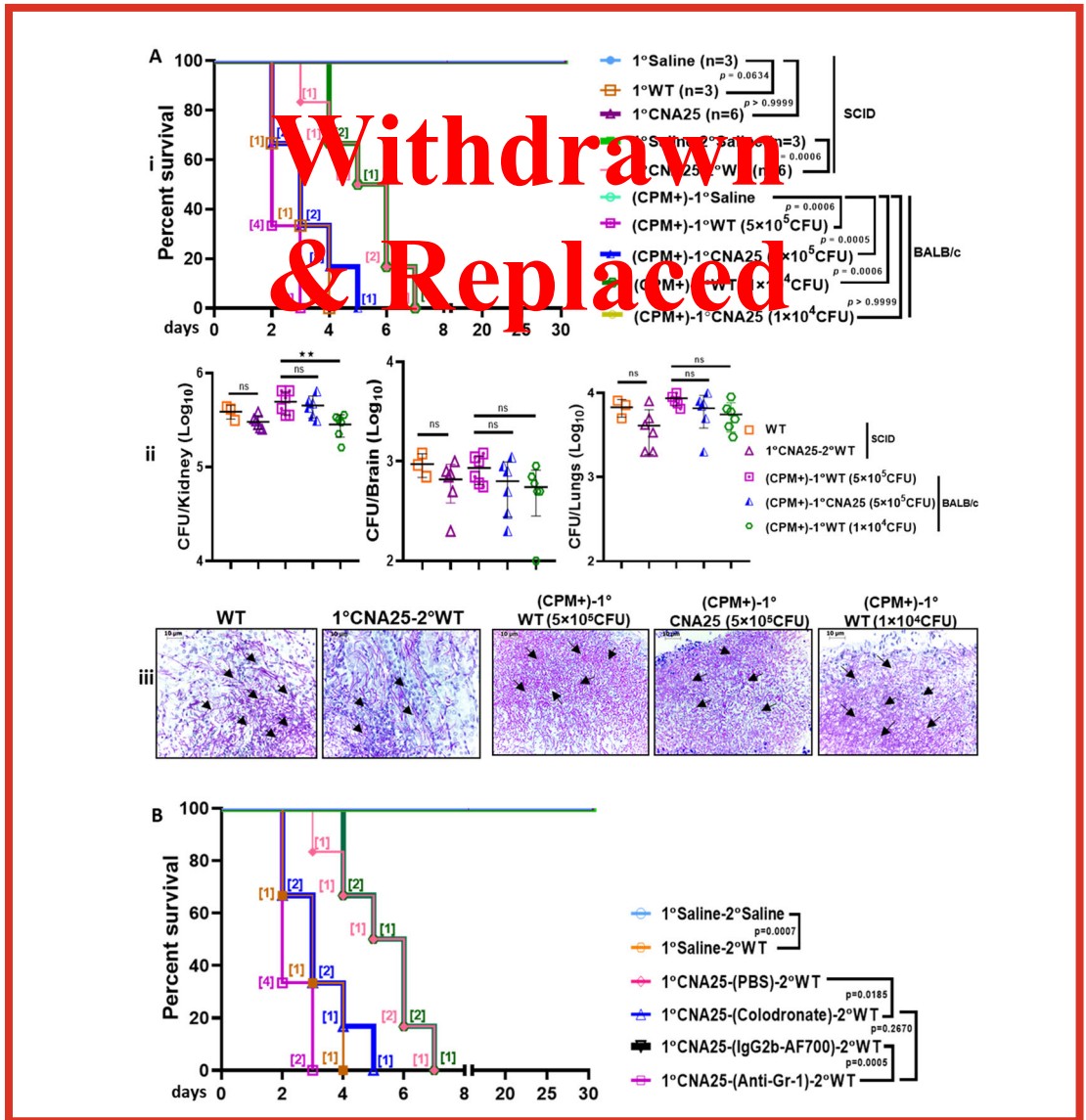

**Figure 9. Response of SCID and CPM-induced immunosuppressed mice towards fungal infection.**

(A) Survival analyses of SCID mice upon intravenous immunization with CNA25 (5 × 10⁵ CFU/mice, $n = 6$), WT *C. albicans* (5 × 10⁵ CFU/mice, $n = 3$), and saline control (100 µl 1× PBS, $n = 3$) was carried out for 30 days. Post 30 days of immunization, CNA25-vaccinated SCID mice were re-challenged with WT (5 × 10⁵ CFU/mice) along with saline controls as denoted as 1°CNA25-2°WT, 1°Saline-2°Saline, respectively. In a similar study, immunosuppression was carried out by CPM (200 mg/kg) administration in naïve and CNA25-vaccinated mice. After 3 days of intraperitoneal CPM treatment, fungal challenges (5 × 10⁵ CFU/mice and 1 × 10⁴ CFU/mice, $n = 6$ each) were carried out and survival was monitored for 30 days. Statistical significance was determined using the log-rank (Mantel–Cox) test. The *P* values are listed beside the corresponding comparison groups (i). Fungal burden in the vital organs of euthanatized mice was determined by CFU analysis (ii) and PAS staining (iii). 1° is primary and 2° denotes re-challenge. Data are the representative of two different sets of experiments, presented as mean ± SEM and analyzed using the Mann–Whitney U test (**$P \leq 0.01$ and ns-nonsignificant). Representative PAS-stained images of the kidney were shown with a 10 µm scale bar. The black arrows indicate the fungal cells. (B) A Kaplan–Meier survival curve was generated to compare the survival rates of Macrophage-depleted (1°CNA25-(Clodronate)-2°WT) and neutrophils-depleted (1°CNA25-(Anti-Gr-1)-2°WT) groups of mice ($n = 6$/group) with their respective PBS and isotype control counterparts (1°CNA25-(PBS)-2°WT and 1°CNA25-(IgG2b-AF700)-2°WT). This experiment was conducted in the presence of saline (1°Saline-2°Saline) and WT lethal control (1°Saline-2°WT). Statistical significance was determined using the log-rank (Mantel–Cox) test. The *P* values are listed beside the corresponding comparison groups.

naïve immune cells were able to clear CNA25 from the host. However, CNA25-vaccinated SCID mice failed to protect the WT re-challenge although there was a delay in death rate again advocating the significance of B and T cells in fungal protective immunity. The median death time was 2 days as opposed to 5 days in sham immunized mice. For further confirmation, we induced immunosuppressed conditions in mice by administrating cyclophosphamide (CPM). CPM as an immunosuppressive agent has a wide

range of non-specific effects on immune cells such as T cells, macrophages, neutrophils etc. (Ahlmann and Hempel, 2016; Winkelstein, 1973; Zuluaga et al, 2006). To verify the effect of CPM, BALB/c mice were injected intraperitoneally (IP) with one dose of 200 mg/Kg, and after 3 days, blood cells were counted (Appendix Fig. S2H) and found that in such treatment ~70% of neutrophils were depleted indicating a neutropenic environment. Such immunosuppressed models were subjected to fungal

infections by intravenously inoculating two different doses ($5 \times 10^5$ and $1 \times 10^4$ CFU) of WT and CNA25 cells and their survivability was monitored. Notably, while these mice succumbed to both WT and CNA25 at $5 \times 10^5$ inoculum size within 6 days of infection ($P = 0.0005$–$0.00006$), the lower dose inoculation of CNA25 only did not induce any death ($P > 0.9999$) but the same low dose of WT was toxic ($P = 0.0006$). This result justifies the critical role of naïve innate immune cells like neutrophils as the first line of defense against invading pathogens even for the attenuated strain like CNA25. Since some neutrophils and other innate cells were still available post-CPM exposure, those were sufficient enough to clear the lower dose of CNA25 cells, and the mice survived. CFU and PAS staining analyses further confirmed the death of the mice due to excessive fungal load (Fig. 9Aii,iii). Nevertheless, this result suggested that CNA25 is truly an avirulent strain.

## Role of primed neutrophils and macrophages in memory response

Trained innate immune cells upon exposure to a pathogen are reported to be involved in long-term protection from homologous and heterologous pathogenic re-challenge by a mechanism called trained immunity (d'Enfert et al, 2021; Lilly et al, 2021; Lilly et al, 2019; Netea et al, 2015). Since our previous experiment implemented the possible involvement of innate cells in initial fungal clearance and also in memory response, to strengthen it, neutrophils and macrophages were specifically depleted by injecting anti-Gr-1 antibody and clodronate, respectively, to CNA25-immunized mice as depicted in Fig. 5Bi and such mice were re-challenged with WT *C. albicans* (Fig. 9B). Necessary isotypes were also injected to obtain zero depletion controls. While all the immunized mice with reduced levels of neutrophils succumbed to reinfection ($P = 0.0005$), only 50% of mice died upon macrophage depletion ($P = 0.0185$). Isotype antibodies did not affect the survival of immunized mice to lethal re-challenge. Cellular depletion of these cells in CNA25-vaccinated mice was confirmed by flow cytometry (Appendix Fig. S2I,J). The severity of fungal sepsis due to fungal burden in the mice was confirmed by determining the CFU score (Appendix Fig. S4D). This result suggested that the trained innate cells like neutrophils and macrophages play an important role in protective immunity against *C. albicans*. Altogether, our results suggested that innate immunity functions both in early and late immune responses via Dectin-1-β-glucan interaction, while B and T cells mediated adaptive immunity functions to substantiate memory response against fungal infection.

## Discussion

It is a well-known fact that fungal pathogens are a serious public health concern and the available antifungal drugs are partially effective in mitigating fungal diseases, therefore, a safe and effective antifungal vaccine is a dire necessity (Sahu et al, 2022). As live attenuated whole-cell vaccines are very similar to the pathogens that cause disease but decorated with a wide range of antigens, they generate robust long-lasting immune responses. Measles, mumps, rubella (MMR combined vaccine), Rotavirus, Smallpox, Chickenpox, Yellow fever, BCG, etc. are some of the examples of live attenuated vaccines that are effectively being used in various

vaccination programs worldwide (Pollard and Bijker, 2021; Vetter et al, 2018). As an approved antifungal vaccine for human use is still awaited, this study explored CNA25, a DNA replication-defective strain of *C. albicans*, as a live whole-cell vaccine against hematogenously disseminated candidiasis in preclinical models and showed that it enhances all the three layers of immunity: innate, adaptive, and trained collectively to develop a strong memory immune response to prevent reinfection.

CNA25 lacks both the alleles of *POL32*, and the role of Pol32 in DNA replication, chromosomal stability, filamentation, cell wall composition, drug resistance, and systemic candidiasis development has already been reported (Patel et al, 2023). Here we showed that intravenous immunization with live CNA25, mice were protected not only against *C. albicans* but also against other NAC species. Consistent with this report, a recent study suggested that BCG, the vaccine against pulmonary tuberculosis, is also more effective via intravenous route than intradermal and aerosol (Darrah et al, 2020). Multiple oral gavage feeding of CNA25 was another effective way of immunization that provided decent protection against the lethal challenge of *C. albicans*. This suggested that while CNA25 is avirulent, it is immunogenic enough to elicit a strong protective immune response to prevent reinfection. Fungal burden and histopathological kinetic analyses suggested a strong positive correlation between fungal load in vital organs and disease severity within 7 days of infection and is one of the critical parameters that induced death in the infectants. A heavy fungal load causes an accumulation of factors like adhesins, tissue-degrading enzymes, candidalysin toxin, etc. to induce tissue damage (Wilson et al, 2016).

Innate immune cells are the front runners in response to a pathogen. The fungal clearance is mediated by 4 key steps; (a) infiltration or accumulation of phagocytes at the site of infection; (b) detection of pathogen-associated molecular patterns (PAMPs) located on the fungal surface via pattern recognition receptors (PRRs) of immune cells; (c) fungal uptake, and (d) killing of phagocytosed cells by lysosomal- mediated phagolysosome (Brown, 2011). The whole blood cell profiling suggested that while the CNA25 infection induced innate immune response showing proliferation peaks by day 3 and contraction by day 7 possibly to restore immune cell homeostasis and such immune response was able to clear fungal cells and increase survivability, pathogenic WT infection in mice could not induce lymphocyte response at least in the peripheral blood. High cell numbers of monocytes and granulocytes by day 7 in WT-infected mice and their association with high fungal load indicated a possible development of a high degree of inflammatory state and disease progression in mice. Analyses of circulating and tissue-resident immune cells profiling showed that the macrophages + DCs population remained high in the mice infected with fungal strain irrespective of their virulence status suggesting their role in early antifungal response. Similarly, the increased population of neutrophils and NK cells mostly in CNA25 strain infected mice indicated their involvement in fungal clearance and protective immunity. As has been reported although macrophages are more efficient in fungal killing than DCs, DCs also play an additional important role in processing and presenting fungal antigens for the activation of T-cell responses (Netea et al, 2004). Cell wall components of fungal pathogens that function as PAMPs play an important role in immune cell recognition. The masking of 1,3-β-glucan on the cell surface prevents phagocytic

recognition and enhances the immune evasion of *C. albicans* (Ballou et al, 2016; Pradhan et al, 2018). Since the thick cell wall of CNA25 possessed an abundant amount of β-glucan and chitin than in WT *C. albicans* cells (Patel et al, 2023), they were unlikely to be concealed, thus were easily recognized and cleared faster than the WT cells by the phagocytes both *in vitro* and *in vivo*. A higher fungal load was always associated with early expression of PRRs like TLR2, TLR4, and MYD88 as observed in WT-challenged mice, while a steady increase of Dectin-1/TLR2 expressions was specific to CNA25. The Dectin-1-β-glucan interaction triggers the Card9-Syk pathway that leads to further activation of NFκB and release of cytokines and chemokines (Drummond and Brown, 2011). Dectin-1 also stimulates phagocytosis and inflammasome activation (Swidergall, 2019). Dectin-1 also collaborates with TLR2 to induce intracellular signaling via SYK- and RAF1-dependent pathways (Dennehy et al, 2008). However, activation of Card9-Syk and RAF1-dependent pathways involvement in immune activation in CNA25-vaccinated mice requires further investigation. Interestingly, blocking of the Dectin-1 receptor by anti-Dectin-1 antibody completely inhibited protection to reinfection in vaccinated mice, thus the fungal clearance and protective immunity induced upon CNA25 vaccination is most likely β-glucan-Dectin-1 interaction dependent.

B and T cells induced immune responses are two arms of adaptive immunity. The increased population of B, B1, and T cells mostly in CNA25-infected mice, and the inability to tolerate lethal re-challenge by the vaccinated SCID mice indicated involvement of adaptive immune responses in antifungal resistance. While the CD4+ T cells are MHC-II-Restricted and pre-programmed for helper functions, CD8+ T cells are MHC-I-restricted and required for cytotoxic functions (Xiong and Bosselut, 2012). Both of these T-cell subtypes were elevated in the CNA25-immunized protected group (1°CNA25-2°WT) and cellular depletion of these cells in the CNA25-vaccinated mice reduced the protection emphasizing the critical function of T cells in protective immunity. Among the CD4+ T cells, IFNγ, TNFα, and IL-17-producing CD4+ T-cell populations were elevated in the protected mice, and the critical role of such T cells in protective antifungal immunity has already been established (Mercante et al, 2006). More importantly, studies suggested that HIV+ persons are vulnerable to developing oropharyngeal candidiasis (OPC) which is associated with a low abundance of CD4+ T cells and our finding further strengthens the significance of CD4+ T cells in fungal immunity (Mercante et al, 2006; Nielsen et al, 1994). Several other studies have demonstrated the role B cells mediated humoral response in fungal immunity (Magliani et al, 1997; Rudkin et al, 2018).

Pro-inflammatory cytokines are primarily responsible for initiating an effective defense against exogenous pathogens. However, overproduction of these mediators can be harmful and may ultimately lead to shock, multiple organ failure, and death. In contrast, anti-inflammatory cytokines are crucial for downregulating the exacerbated inflammatory process and maintaining homeostasis for the proper functioning of vital organs, but an excessive anti-inflammatory response may also result in the suppression of the body's immune function. The increase in both pro- and anti-inflammatory cytokines levels in fungal-infected mice in an early hour (3 h) to 7 days post infection suggested early activation of immune responses to both pathogenic and non-pathogenic strains, although the levels were much higher in WT than in CNA25 and

re-challenged group. Accordingly, while a very high level of IL-1, IL-2, IL-6, IL-12, IL-17, IFNγ, and TNFα could induce high inflammation, a high level of anti-inflammatory cytokines such as IL-13, IL-10, IL-9. IL-5, IL-4, and IL-3 could attenuate the anti-*Candida* response in the WT-infected group, and a combination of both allows the fungal cells to thrive and induce tissue damage and death within 8 days of WT infection. However, an optimal balance between pro- and anti-inflammatory cytokines-mediated responses in CNA25-challenged mice resulted in anti-candidal defense as the molecular ablation of IL-17, IFNγ, and TNFα in CNA25-vaccinated mice caused death only when they were re-challenged with WT *C. albicans*. IL-10 is the cytokine released from macrophages and DCs to block the production of other cytokines from Th-1 cells and remained at a very high level on the 7th day in the WT-infected mice, suggesting an onset of immunosuppression resulting in death in later days. Cytokines IFN-γ, IL-2, IL-12, and IL-18 help in the activation of macrophages and nitric oxide production, generation of cytotoxic T cells, production of opsonizing antibodies, and delayed-type hypersensitivity (Netea et al, 2015). Consistent with our data, IFNγ knockout mice are highly susceptible to disseminated *C. albicans* infection, and recombinant IFNγ treatment is reported to protect them against systemic candidiasis (Balish et al, 1998; Kullberg et al, 1993; Shalaby et al, 1985). Th17-response results in the production of IL-17, IL-21, and IL-22 cytokines. IL-17 has been shown to upregulate many chemokines and matrix metalloproteases through the NFκB and MAPK signaling pathways, leading to the recruitment of neutrophils into the sites of inflammation and playing a critical role in host defense. Accordingly, IL-17R null mice are highly susceptible to systemic candidiasis and we also observed that IL-17 depletion resulting death in CNA25 mice due to fungal reinfection (Huang et al, 2004). This study also reported a direct role of pro-inflammatory cytokine TNFα in fungal clearance. Chemokines are released at an early stage of fungal infection and bind to their respective receptors to trigger interleukin production to promote fungal clearance (Traynor and Huffnagle, 2001). We reported higher levels of CC-type chemokines (MCP-1, MIP-1a, MIP-1b, RANTES, and eotaxin) in CNA25-immunized mice. Platelets are known to produce some of these immune mediators like RANTES and PF4 with antimicrobial activity against *Candida* species, and platelet-rich plasma is reported to inhibit the growth of *Candida* (Drago et al, 2013) and we observed a positive correlation with fungal load and platelet counts in fungal-infected mice as a counteractive mechanism. More importantly, our data obtained from the cytokine depletion experiments has significant implications, particularly in understanding the roles of these cytokines in vaccine-induced protection upon reinfection.

In addition to B- and T-cell function, innate cells are found to be involved in developing immunological memory to protect against reinfection (Netea et al, 2015). In trained immunity, epigenetic reprogramming of neutrophils and monocytes induced by β-glucan of *C. albicans* enhances cytokines production and fungal clearance. The training requires Dectin-1-β-glucan interaction and activation of Raf-1pathway (Netea et al, 2015). Since Dectin-1-β-glucan interaction seems to be actively operational in CNA25-vaccinated mice, it was intriguing to test the concept of adaptive immunity by trained neutrophils and macrophages. Depletion of these cells by anti-Gr-1 antibody and clodronate, respectively, in CNA25-immunized mice ascertained the role of primed innate immune

cells in memory response as all the mice succumbed to re-challenge. Although we have not explored the role of NK cells in memory immune response against fungal infection, they might also play a role even in trained immunity. However, our data do not distinguish between the two possibilities of trained innate vs. adaptive immune-dependent myeloid priming upon reinfection in inducing protective immunity and it further requires a more elaborate investigation.

Vaccines based on live attenuated pathogens do not need adjuvants and as they are multi-antigenic effectively activate several components of the immune system at once. The current results suggested that CNA25 is such an avirulent strain that is highly immunogenic and induces various layers of immune responses such as innate, adaptive, and trained to protect the hosts from various fungal infections. Thus, CNA25 is a potential live whole-cell vaccine candidate to be explored in further trials.

# Methods

## Ethical statement

The Animal Ethical Committee, Institute of Life Sciences, Bhubaneswar, India, reviewed and approved the animal's usage and related protocols, and an ethical Permit Number ILS/IAEC/133-AH/AUG-18 was assigned. Based on the committee's recommendation, all the animal experiments were conducted. Every effort was made to minimize animal suffering and ensure the highest ethical and humane standards.

## Mice, cell line, fungal strains, oligonucleotides, and reagents

The animal house breeds BALB/c (male/female) mice of 6-to-8-week were kept in individually tagged well-ventilated cages at a temperature of $22 \pm 1\,°C$, relative humidity of $55 \pm 10\%$, and a light/dark cycle of 12 h/12 h, with free access to food and water. The SCID mice (NOD.CB17-Prkdcscid/NCrCrl-male) were purchased from Vivo Bio Tech Ltd, Hyderabad, India, and housed with intrinsic care. The RAW 264.7 murine macrophage cell line was obtained from the National Centre for Cell Science (NCCS), Pune, India. The WT *C. albicans* SC5314 and its derivative patented strain CNA25, *C. glabrata* CBS138, *C. tropicalis* ATCC-750, and *C. parapsilosis* ATCC-22019 were used. The periodic acid-Schiff (PAS) stain kit (mucin stain; ab150680) was obtained from Abcam, Cambridge, MA, USA. The Dulbecco's Phosphate Buffered Salt Solution without Ca and Mg (Cat# P04-36500) and Fetal Bovine Serum-South American Origin (Cat# P30-3302) were procured from PAN Biotech, and RPMI 1640 media (Cat# 11875093) was obtained from Thermo Fisher Scientific. Ammonium chloride ($NH_4Cl$), sodium bicarbonate ($NaHCO_3$), and disodium ethylene diamine tetra acetate dehydrate ($Na_2$-EDTA) for RBC lysis buffer were purchased from Sigma-Aldrich. The LIVE/DEAD™ Fixable Blue Dead Cell Stain Kit (Cat# 2089943) and the Zombie Yellow™ Fixable Viability Kit (Cat# 423103) were purchased from Thermo Fisher Scientific Inc, MA, USA, and BioLegend, respectively. The antibodies and oligonucleotides are listed in Tables 1 and 2.

## Phagocytosis and fungal clearance assay

The fungal uptake and killing assay was conducted as described before (Kumari et al, 2023). Briefly, the CFSE-stained fungal cells and deep-red-stained RAW murine macrophages (ATCC-TIB71™) were mixed in a 1:1 ratio ($0.5 \times 10^6$ cells) and co-cultured for different time points (1, 2, and 3 h). After every time interval, the supernatant was discarded, and the adhered cells were scrapped and pooled for flow cytometry acquisition. Cells were acquired in BD Fortessa and double-positive populations were segregated using the FITC channel (CFSE) and APC channel (deep red). The phagocytosis efficiency was determined by gating the cells and the percentage of double-positive cells was calculated. A similar experiment was carried out by taking unstained RAW 264.7 murine macrophage cells and *C. albicans* cells for fungal clearance assay by CFU analysis. At every time interval, co-cultures were washed to exclude non-adhered cells, and 1 ml of warm water (50–60 °C) was added to each well to release the phagocytosed *C. albicans*. The cells from the wells were scrapped, collected, and diluted to a desired dilution for spreading in YPD agar plates. Plates were incubated at 30 °C for 48 h, colonies were counted and plotted as bar graphs using GraphPad Prism 8. For macrophage killing assay, the co-culture of RAW 264.7 cells with WT or CNA25 *C. albicans* cells was stained with Propidium iodide, and fluorescence imaging was carried out using EVOS imaging system, Thermo Fisher Scientific to determine the lysed macrophage population. Counted dead macrophage cells were plotted.

## Immunization and systemic candidiasis development assay

The fungal inoculum for various fungal infections was conducted by using standard protocol (Bose et al, 2023; Kumari et al, 2023; Patel et al, 2023; Peroumal et al, 2019). The live and heat-killed (HK) CNA25 inoculums were prepared from the same pre-culture. The overnight grown *C. albicans* culture was washed thoroughly with sterile distilled water followed by washing with PBS and finally dissolved in PBS. The cells were counted using the Neubauer chamber slide in the central area ($5 \times 5$ squares) and $5 \times 10^6$ cells/ml cell suspension was prepared. For the heat-killed inoculum, $5 \times 10^6$ cells/ml cell suspension was heated at 70 °C for 30 min. The complete killing and counting of the cells were verified by spreading 100 μl of live or heated cells on YPD+agar plates and incubated further at 30 °C overnight. Groups of BALB/c mice ($n = 8$/group) were intravenously immunized via lateral vein with a dose of $5 \times 10^5$ CFU of live or heat-killed CNA25 per mouse (100 μL of $5 \times 10^6$ cells/ml). Sham immunization was carried out with saline control. After 30 days of immunization, $5 \times 10^5$ CFU WT *C. albicans* was intravenously challenged, and mice survivability was monitored. To determine the suitable route of immunization, groups of BALB/c mice ($n = 8$/group) were infected with the same CFU of CNA25 via subcutaneous (SC) and intraperitoneal (IP). Similarly, a set of mice was allowed to oral gavage CNA25 for a week. After 30 days, a secondary (2°) lethal challenge via lateral vein with WT *C. albicans* was carried out and survival was monitored. All the challenged mice were carefully monitored for the development of a moribund state and based on humane endpoints such as being lifeless, disinterested in food or water, curved body posture, and nonreactive to finger probing, they

**Table 1. List of antibodies used in the study for immunophenotyping and cellular and molecular depletion assays.**

| Fluorophore | Marker | Isotype | Company | Cat# | Dilutions used for flow cytometry |
|---|---|---|---|---|---|
| BV510 | CD43 | Rat DA x LOU IgG2a, κ | BD Horizon | 563206 | 1:100 |
| Alexa Fluor 488 | CD5 | Rat IgG2a, κ | Biolegend | 100612 | 1:200 |
| PerCp/Cyanine 5.5 | CD19 | Rat IgG2a, κ | TONBO Biosciences | 65-0193-U100 | 1:100 |
| APC | CD161 | Mouse IgG2a, κ | TONBO Biosciences | 20-5941-U100 | 1:100 |
| Alexa Fluor 700 | Gr-1 | Rat IgG2b, κ | Biolegend | 108422 | 1:200 |
| APC-Cyanine 7 | CD11b | Rat IgG2b, κ | TONBO Biosciences | 25-0112-U100 | 1:100 |
| PE | CD4 | Rat IgG2b, κ | TONBO Biosciences | 50-0042-U100 | 1:100 |
| Alexa Fluor 488 | CD8 | Rat/IgG2a, k | Invitrogen | 53-0081-82 | 1:100 |
| APC | CTLA4 | Armenian Hamster IgG | TONBO Biosciences | 20-1522-U100 | 1:100 |
| Alexa Fluor 488 | IFNγ | Rat IgG1, κ | Invitrogen | 53-7311-82 | 1:200 |
| Alexa Fluor 700 | TNFα | Rat IgG1 | Biolegend | 506338 | 1:200 |
| PerCp Cyanine 5.5 | IL-17 | Rat IgG2a, k | Invitrogen | 45-7177-82 | 1:100 |
| Unconjugated | GK1.5 antimouse CD4 | – | InVivoPlus +, Bio X cell | BP0003-1 | NA |
| Unconjugated | – | Rat IgG2b isotype control;anti-keyhole limpet hemocyanin | InVivoPlus +, Bio X cell | BP0090 | NA |
| Unconjugated | Fc-hDectin-1a | – | InvivoGen | Fc-hdec1a | NA |
| Conjugated | – | Goat antihuman IgG secondary antibody Alexa Fluor 594 plus | Invitrogen | A48278 | NA |
| Unconjugated | TLR2 | – | Invitrogen | MA5-16200 | NA |
| Conjugated | – | Mouse IgG2a FITC | Invitrogen | 11-4724-42 | NA |
| Conjugated | – | Mouse IgG secondary antibody Alexa Fluor 488 | Invitrogen | A-11017 | NA |
| Conjugated | – | Rat IgG2a κ isotype FITC | Invitrogen | 11-4321-80 | NA |
| Conjugated | – | Rat IgG1 κ isotype Alexa Fluor 488 | Invitrogen | 11-4301-82 | NA |
| Conjugated | – | Rat IgG1 κ isotype TNFα Alexa Fluor 700 | Biolegend | 400420 | NA |
| Conjugated | – | Rat IgG2a κ isotype PerCp-Cyanine 5.5 | BD Biosciences | 550765 | NA |
| Conjugated | – | Rat IgG2b κ isotype Alexa Fluor 700 | Invitrogen | 56-4031-80 | NA |

*NA* not applicable.

**Table 2. List of oligonucleotides used in RT-PCR analysis.**

| Target | Primers | Sequence (3'-5') |
|---|---|---|
| *DECTIN-1* | *DECTIN-1* F | CCAGCTAGGTGCTCATCTACTG |
| | *DECTIN-1* R | CCTTCACTCTGATTGCGGGAAAG |
| *TLR2* | *TLR2* F | TGCCCAGATGGCTAGTGG |
| | *TLR2* R | CAGAAACTATGATTGCGGACAC |
| *TLR4* | *TLR1* F | GGACTCTGATCATGGCACTG |
| | *TLR1* R | CTGATCCATGCATTGGTAGGT |
| *MYD88* | *MYD88* F | TGACTTCCAGACCAAGTTTGC |
| | *MYD88* R | GAATCAGTCGCTTCTGTTGGA |

were euthanized. This time point was considered as the death point of a given animal and the mice were dissected and vital organs (Kidney, Lungs, and Spleen) were collected for further analyses. The survival curve was plotted using GraphPad Prism 8 software for each group of mice. The surviving mice beyond 30 days were allowed to survive for another 6 months and then handed over to the animal facility for necessary disposal unless mentioned otherwise.

## Immunization to immune-compromised mice

SCID and cyclophosphamide-induced immunosuppressed model of BALB/c mice were used. A group of SCID mice was intravenously challenged with 100 μl of saline ($n = 3$), $5 \times 10^5$ CFU of WT ($n = 3$), and $5 \times 10^5$ CFU of CNA25 ($n = 6$), and their survival was monitored. After 30 days, the primary-challenged (1°CNA25) survived SCID mice were again re-challenged with a lethal dose of WT ($5 \times 10^5$ CFU) and were observed. To develop the immuno-suppressed model, the body weight of BALB/c male mice was determined and grouped into separate cages with six mice each. Cyclophosphamide (CPM) was dissolved in PBS and intraperito-neally (IP) injected into BALB/c mice at 200 mg/kg. On the 3rd day, WT and CNA25 were intravenously (IV) injected with two different doses, $5 \times 10^5$ CFU and $1 \times 10^4$ CFU, in individually caged mice along with saline control, and survival was monitored. A similar experiment was conducted to measure neutrophil depletion upon CPM treatment by flow cytometry. The euthanized mice were dissected and vital organs were collected for further

analyses. The survival curve was plotted using GraphPad Prism 8 software for each group of mice.

## Systemic candidiasis by NAC species and cross-species protection assay

To determine the lethal dose of fungal inoculum, a group of BALB/c mice ($n = 8$/group) was infected intravenously with three different NAC species (*C. glabrata*, *C. tropicalis*, and *C. parapsilosis*) along with saline control and survival was monitored for 30 days post infection. Two doses ($5 \times 10^5$ and $6 \times 10^6$ CFU/mouse) were injected for *C. tropicalis* and *C. parapsilosis*, whereas $5 \times 10^5$ and $1 \times 10^7$ CFU per mouse of *C. glabrata* were administered. For cross-species protection, after 30 days post-immunization with CNA25 ($5 \times 10^5$ CFU/mouse) intravenously, BALB/c mice were re-challenged with *C. tropicalis* ($6 \times 10^6$ CFU), *C. parapsilosis* ($6 \times 10^6$ CFU) and *C. glabrata* ($1 \times 10^7$ CFU). The vital organs were collected from the mice that were succumbed for further analyses. The survival curve was plotted using GraphPad Prism 8 software for each group of mice.

## Kinetic analyses in primary and post-immunized mice

A full-fledged experimental module was designed to examine fungal load, blood profile, and the overall immune response of mice towards infection and immunization to correlate fungal dissemination and clearance in the mice model. The whole experiment was carried out in two parts pre-immunization (1°) and post-immunization (1° and 2°). In the pre-immunization study, a group of mice was intravenously injected with saline, $5 \times 10^5$ CFU/mice of WT and CNA25 *C. albicans*. In the post-immunization study, BALB/c mice ($n = 40$) were immunized intravenously with $5 \times 10^5$ CFU/mice CNA25 along with a similar group of sham immunization with saline. After one month of immunization, while the CNA25-immunized mice were intravenously challenged with $5 \times 10^5$ CFU/mice of WT *C. albicans*, the sham immunized mice were again injected with saline. At different time points (3 h, 3 days, 7 days, 15 days, and 30 days) mice ($n = 8$ from each group of pre- and post-immunization studies) were euthanized and vital organs were collected for various analyses. Since the mice primarily challenged with WT *C. albicans* rarely survive beyond 7 days, their kinetic analyses were not followed after this time point. Blood was collected in two tubes from each mouse just prior to their sacrifice by cardiac puncture. About 200 μl blood was kept in a tube with $K_2$EDTA as an anticoagulant (for CBC count-50 μl and WBC isolation-150 μl) in a 1:10 ratio, while ~300 μl blood was kept without anticoagulant for serum isolation. The serum was collected by centrifuging the blood (without $K_2$EDTA) at $1000 \times g$ for 10 min at 4 °C and stored at −80 °C in individually labeled cryovials until further analysis. The individual spleens were collected in complete RPMI media and kept in ice immediately for splenic cell isolation. The kidneys were harvested, out of 16 kidneys from each mice group, 8 kidneys were kept in $1\times$ PBS for CFU assay. The remaining eight kidneys were dissected longitudinally, while one half was kept at −80 °C with 300 μL TRIzol reagent for RNA isolation, and the other half was kept in 4% Formalin for tissue histology. Other vital tissues like the lungs and brain were also collected in PBS for CFU assay.

## CFU assay

To determine the fungal burden in various tissues such as the kidney, lung, and brain, first they were homogenized individually with a tissue homogenizer and then serial dilution was prepared with $1\times$ PBS up to $10^{-3}$. While the $10^{-2}$ and $10^{-3}$ dilutions of kidney homogenate were spread, $10^{-1}$ and $10^{-2}$ dilutions of lungs and brain were spread on multiple chloramphenicol (34 μg/ml) containing YPD+agar plates to get isolated colonies. Plates were kept at 30 °C for 2 days. Colonies were counted for all dilutions, and back-calculated based on the dilution factor, average data of similar dilution plates was only considered and plotted graphs in a GraphPad prism 8 for individual organs of a specific group and duration.

## Kidney PAS staining

Formalin-fixed kidney samples were sectioned in LEICA RM2125 RTS microtome with 5 μm thickness, dewaxed in xylene, and rehydrated gradually in decreasing concentration of ethanol (100%, 95%, and 75%) and distilled water. Sections were stained with 0.5% periodic acid solution for 5 min, rinsed with distilled water, and dipped in Schiff reagent for 15 min followed by counterstaining with Mayer's hematoxylin for 1 min then washed with lukewarm tap water. The slides were dehydrated gradually with increasing concentrations of ethanol (75%, 95%, and 100%), then mounted with D.P.X. mounting liquid, and images were acquired under a LEICA MD500 microscope with $\times 40$ magnification.

## Gene expression analysis by RT-PCR

Kidneys were homogenized gently with a sterile micro-homogenizer and total RNA was isolated using the Gene JET RNA purification kit as per the manufacturer's protocol. About 2 μg of RNA was used to synthesize cDNA using a high-capacity cDNA reverse transcription kit with the provided random primers, and quantitative RT-PCR was performed using specific sets of primers. The PCRs were set in a volume of 20 μl with 100 ng of cDNA, 10 pmol primers mix, and 10 μl of SYBR Green master mix. The qRT-PCR was carried out in Quant Studio 3 with fast cycle conditions having initial denaturation at 95 °C for 2 min followed by 40 cycles of 95 °C for 5 s denaturation and 60 °C for 30 s for annealing and extension. The data obtained were analyzed using the $2^{-\Delta\Delta CT}$ method. The gene expression in WT and CNA25 injected kidneys was normalized with *HPRT1*, and graphs were plotted using GraphPad Prism 8 software for individual genes with different time points. The experiment was carried out twice and the average Ct value of the same gene was considered for fold change calculation.

## Complete blood profiling of mice

Within six hours of the blood collection, the complete blood count (CBC) analysis was carried out by taking 20 μl of anticoagulant blood and examining it in the Exigo Veterinary Hematology Blood analyzer. The data sheet provides the list of parameters analyzed with the observed value alongside the normal values. The data was exported in the image format and individual graphs were plotted for each blood cell with different time points in the GraphPad Prism 8. The experiment was repeated twice.

## WBC immunophenotyping

About 150 μl of blood with $K_2$EDTA was transferred into a 15 ml tube, and 2 ml 1 × RBC lysis buffer was added and incubated for 2–3 min on ice. After lysis, the sample was neutralized by adding 1 ml of PBS + 0.1% BSA. The sample was centrifuged at 380 × $g$ for 5 min at 4 °C, the supernatant was discarded, and cells were again washed with 1 ml of neutralization buffer. Cells were resuspended in 500 μl of PBS + 0.1% BSA, counted, and ~1 million cells were transferred into properly labeled sterile tubes for flow cytometry. Then the live–dead staining was done using the Fixable Blue Dead Cell Stain Kit. Briefly, 1 ml of PBS was added to an individual tube containing 1 million cells followed by 1 μl of solution A from the kit, and the cells were incubated for 15-20 min in the dark at room temperature. After incubation, cells were centrifuged at 380 × $g$ for 5 min at 4 °C and resuspended in 500 μl of PBS + 0.1% BSA. Antibody staining was carried out with the following fluorescent-tagged antibodies APC-Cy7 anti-mouse CD11b, Alexa flour 488 anti-mouse CD5, PerCP-Cy5.5 anti-mouse CD19, and Alexa flour 700 anti-mouse Gr-1 (Ly6G/Ly6C). Antibodies were added individually, and cells were incubated for 30 min in the dark at room temperature, followed by PBS + 0.1% BSA washing and resuspension in PBS + 0.1% BSA. For better discrimination among the population, unstained and single antibody stain controls were used, along with samples. The samples were acquired using a BD LSRFortessa Flow Cytometer (BD Biosciences), compensation was done post-acquisition to prevent spectral overlap, FCS files were exported, and data were analyzed using FlowJo v8.2.0 software. The percentage of cells was plotted in GraphPad prism.

## Murine splenocytes isolation and phenotyping

For the single-cell suspension, the spleens were processed in the cell culture-based sterilized vertical airflow air safety cabinet. Spleens were washed with PBS + 0.1% BSA, residual fat was removed from the spleen, and placed in a cell strainer kept in a sterilized Petri dish having 2 ml of PBS containing 0.1% BSA. The spleen was meshed gently with the cell strainer into the Petri dish using the plunger end of sterilized syringes individually. For each spleen, a separate set of cell strainer, plunger, and Petri dish was used. The cell strainer was properly rinsed with 5 ml PBS + 0.1% BSA and suspended cells were collected carefully from the Petri dish individually in a 15-ml falcon tube. The suspended cells were centrifuged at 380 × $g$ for 5 min at 4 °C. After centrifugation, the supernatant was discarded and cells were suspended with 2 ml of 1 × RBC lysis buffer, mixed by gentle inversion, and incubated for 5–10 min in ice. Post-incubation suspension volume was maintained up to 13 ml each with the same RBC Lysis buffer and centrifuged. The RBC lysis process was performed twice. After RBC lysis, cells were resuspended in 1 ml of PBS + 0.1%BSA, centrifuged, the supernatant was discarded, and the cells were resuspended in 1 ml PBS + 0.1%BSA. About 10 μl of single-cell suspension was mixed with trypan blue, counted in a hemocytometer, and kept in ice until staining.

Splenic cells were transferred to respective tubes with a final volume of 1 ml in two separate sets of 1.5 million cells, each for two different groups of antibody staining. The cells were washed with 1 ml of PBS + 0.1% BSA followed by antibody staining. The first set of staining was carried out with APC-Cy7 anti-mouse CD11b, Alexa

flour 488 anti-mouse CD5, PerCP-Cy5.5 anti-mouse CD19, Alexa flour 700 anti-mouse Gr-1 (Ly6G/Ly6C), APC anti-mouse NK1.1 (CD161) and brilliant violet 510 anti-mouse CD43 antibodies to determine the Myeloid (Neutrophils, Macrophage + DCs) and Lymphoid Population (B cells, T cells, NK cells, and CD43 + NK cells). The second set of staining was carried out with PE anti-mouse CD4, Alexa fluor 700 anti-mouse CD44, and Alexa Fluor 488 anti-mouse CD8a antibodies to discriminate T-cell subpopulations ($CD4^+$ and $CD8^+$ T cells). Each antibody was added individually in all the samples along with unstained control, and single-colored control, and again incubated for 30 min in the dark at room temperature for optimal staining. After incubation, cells were washed with PBS + 0.1% BSA, and live–dead stinging was performed using the Fixable Blue Dead Cell Stain Kit as discussed earlier. The samples were acquired using a BD LSRFortessa Flow Cytometer (BD Biosciences), compensation was done to prevent spectral overlap, FCS files were exported, and data were analyzed using FlowJo 8.0 software. The percentage of cells was plotted in GraphPad prism.

## $CD4^+$ T cells isolation and determination of various subpopulations

Approximately $5 \times 10^7$ splenocytes were used for the $CD4^+$ T Cells isolation. The $CD4^+$ T cells were isolated using the Dynabeads™ Untouched™ Mouse CD4 Cells Kit based on the manufacturer protocol. Untouched CD4 + T cells were activated with phorbol 12-myristate 13-acetate (PMA) at a concentration of 10 ng/ml and Ionomycin at a concentration of 1 μg/ml for 4 h under the standard aseptic cell culture condition in RPMI 1640 complete medium, at the 3rd hour of this incubation 1 μl of brefeldin A (3 ng/ml) was added. Finally, the cells were harvested and washed twice in 2% FBS, and 2 mM EDTA in phosphate-buffered saline pH 7.4, and live–dead blue staining was done followed by APC anti-mouse CTLA4. After staining cells were washed and fixed in 150 μl BD Cytofix/Cytoperm, fixation/permeabilization solution. The fixed cells were washed and permeabilized in 400 μl of BD Perm/Wash Buffer, stained with fluorescently tagged antibodies such as Alexa flour 488 anti-mouse Interferon-gamma, Alexa flour 700 anti-mouse tumor necrosis factor-alpha, and PerCPcy5.5 Interleukin-17A antibodies for the determination of IFNγ, TNFα, and IL-17-positive population. After washing twice, the stained cells were acquired in BD LSRFortessa and analyzed using Flowjo8.0 Software.

## Serum cytokine estimation

Once all the time point experiments were over, the serum stored at −80 °C was taken out and cytokine profiling was determined by using a Bio-Plex Pro Mouse Cytokine 23-plex Assay kit. In brief, serum samples were diluted with sample diluent in a 1:4 ratio and similarly eight standards (S1 to S8) samples were prepared by fourfold dilution with standard diluent available in the kit. The mouse cytokine 1 × assay bead was added to a 96-well transparent flat-bottom black plate and washed with 100 μl wash buffer. About 50 μl of each sample along with standards and blank control (Only assay buffer) were added in the respective wells of 96-well plates with proper labeling in the plate template sheet and incubated for 30 min in a plate shaker with 850 + /− 50 rpm at 25 °C followed by washing with 100 μl wash buffer. Then 25 μl of 1 × detection antibody was added in each well and incubated for 30 min in the dark by covering with aluminum foil and again washed

with 100 µl of wash buffer. The $1 \times$ SA-PE (streptavidin-phycoerythrin) (50 µl) was added in each well and incubated for 20 min in the dark followed by washing and resuspension in 100 µl of assay buffer. The plate was covered and mixed in a plate shaker with $850 + /- 50$ rpm at 25 °C for 1 min and the plate was analyzed in Bio-Plex® 200 System (Bio-Rad). A total of 20 different cytokines were analyzed namely; IL-1α, IL-1β, IL-2, IL-3, IL-4, IL-5, IL-6, IL-9, IL-10, IL-12 (p40), IL-12 (p70), IL-13, IL-17, IFN-γ, TNF-α, G-CSF, KC, MCP-1, MIP-1α, MIP-1β, and RANTES, data were exported, and graphs were platted for each cytokine with individual groups and different time points in picogram/ml. The experiment was repeated twice.

## In vivo molecular blocking and cellular depletion assay

The selective blocking or depletion of specific molecules and cell types was achieved using monoclonal antibodies targeting the desired populations as described before (Lilly et al, 2021; Lilly et al, 2019). Briefly, BALB/c mice ($n = 6$/group) were intravenously (IV) immunized with CNA25 ($5 \times 10^5$ CFU), along with a saline control denoted as 1°CNA25 and 1°Saline groups. On the 15th day post-immunization, the mice were intraperitoneally (IP) injected with 100 µl of specific antibodies such as Fch-hDectin-1a (25 µg/kg for Dectin-1 blocking), anti-mouse TLR2 (1 mg/kg for TLR2 blocking), anti-mouse IFNγ (1 mg/kg XMG1.2 for IFNγ blocking), anti-mouse TNFα (1 mg/kg for TNFα blocking), anti-mouse IL-17 (0.5 mg/kg for IL-17 blocking), anti-mouse CD4 (GK1.5) (10 mg/kg for CD4$^+$ T lymphocytes depletion), anti-mouse CD8 (1 mg/kg for CD8$^+$ T lymphocytes depletion), anti-mouse Gr-1 (1 mg/kg Ly6G/Ly6C for neutrophils depletion), and respective isotype controls with specific fluorochrome conjugation (rat IgG2b anti-keyhole limpet hemocyanin (10 mg/kg), rat IgG2a (0.5–1 mg/kg), rat IgG1 (1 mg/kg), rat IgG2b (1 mg/kg)). For macrophage depletion, 100 µl of clodronate (50 mg/kg) or PBS was injected. All the above antibodies and isotypes were diluted in sterile PBS as per the mentioned concentration. After 2 days of antibody administration, $5 \times 10^5$ CFU of WT and 100 µL of saline were injected intravenously. Immediately after 2 h of fungal challenge, another dose of respective antibodies or isotypes were provided intraperitoneally, followed by three such administration at every two-day interval. Survivability was monitored for 30 days. The mice were sacrificed based on the severity of the infection and the humane endpoint criteria. Post-sacrifice, blood and vital organs such as the spleen and kidney were collected to determine depletion efficiency via flow cytometry and fungal load via CFU assay.

### Depletion confirmation by flow cytometry
The blood and splenic cells were processed according to the methods outlined in previous sections on WBC immunophenotyping and splenic phenotyping. About 1 million blood cells were used for determining neutrophil, macrophage, Dectin-1, and TLR2 depletion, while a similar quantity of splenic cells was processed for CD4$^+$ and CD8$^+$ determination. The purified CD4 cells were processed for the evaluation of IFNγ, TNFα, and IL-17 levels. The cells were transferred to appropriately labeled tubes used for flow cytometry, and live–dead staining was performed using Zombie Yellow dye. Subsequently, specific fluorophore-conjugated antibodies were added, and the samples were incubated at 4 °C in the dark for 30 min. Each samples were taken for acquisition after washing along with unstained, live–dead, and single-color controls in BD LSRFortessa Flow Cytometer (BD Biosciences). Compensation was done post-acquisition to prevent spectral overlap, FCS files were exported, and data were analyzed using FlowJo v8.2.0 software. The percentage of cells was plotted in GraphPad prism.

**The paper explained**

**Problem**

Due to a lack of effective antifungal drugs and vaccines, about 1.5 million people per year succumb to fungal infections. This rate is increasing due to the rise in immunocompromised individuals and ever-evolving multi-drug-resistance fungal species. This report explored the implication of an attenuated *Candida albicans* strain as a potential vaccine against pan-fungal infections in preclinical models.

**Results**

CNA25 strain lacks the smallest subunit of DNA polymerase delta and is avirulent. Vaccination with live CNA25 prevents a wide range of fungal infections. T cells and trained innate cells mediated protective immunities are critical to prevent reinfection in vaccinated mice. Dectin-1 receptors and cytokines like IFNγ, TNFα, IL-17, etc., play important roles in pathogen recognition, immune activation, and fungal cell clearance to prevent related deaths.

**Impact**

Since CNA25 can protect the mice from lethal re-challenge infection from pan-fungal species, it is a promising vaccine candidate that can benefit the critical unmet need posed by systemic candidiasis.

## Statistical analysis

All graphs depicting mice survival experiments, flow cytometric analysis, RT-PCR, hematology blood analysis, and Cytokine profiling were generated using GraphPad Prism 8.0 software. Statistical significance in all Kaplan–Meier survival graphs was determined using the log-rank (Mantel–Cox) test. Analysis of blood parameters, RT-PCR data, immune cell counts, and cytokine profiles was conducted through two-way ANOVA (Tukey's multiple comparisons test). Fungal colony-forming unit (CFU) data were assessed using the Mann–Whitney $U$ test, whereas the Chi-square (and Fisher's extract) test was performed in fungal-macrophage cells interaction experiments. Significance levels were indicated by $P$ values as $* \leq 0.05$, $** \leq 0.01$, and $*** \leq 0.001$.

## For more information

For more information about the authors, please refer to https://www.ils.res.in/scientists/narottam-acharya/ and for gene https://www.candidagenome.org/cgi-bin/locus.pl?locus=pol32&organism=C_albicans_SC5314.

## Data availability

This study includes no data deposited in external repositories. The source data of this paper are collected in the following database record: accession number S-BSST1394.

The source data of this paper are collected in the following database record: biostudies:S-SCDT-10_1038-S44321-024-00080-8.

# Peer review information

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

## Acknowledgements

The authors thank Mr. Sitendra Prasad Panda, Mr. Paritosh Nath, Mr. Biswajit Patro, Mr. Subrat Kumar Naik, and Dr. Sarita Jena for technical assistance and nurturing of the animals. The authors thank our other laboratory colleagues for their thoughtful discussion. SRS and BGU are grateful for the SERB/UGC-Senior Research Fellowships and AD and DP are thankful for the DBT-RA fellowships. ILS animal house, TEM imaging, and other central facilities are highly acknowledged. We acknowledge receiving *C. glabrata* CBS138 strain from Dr. R Kaur, CDFD, Hyderabad, and *C. tropicalis* ATCC-750 and *C. parapsilosis* ATCC-22019 strains from Dr. Soma Rohotgi, IIT-Roorkee, Uttarakhand, India. This work was supported by the intramural core grant from ILS and extramural research funds from DBT, India (BT/PR32817/MED/29/1495/2020).

## Author contributions

**Satya Ranjan Sahu**: Resources; Software; Formal analysis; Validation; Investigation; Visualization; Methodology; Writing—original draft; Writing—review and editing. **Abinash Dutta**: Resources; Software; Formal analysis; Validation; Investigation; Visualization; Methodology; Writing—original draft; Writing—review and editing. **Doureradjou Peroumal**: Software; Formal analysis; Investigation; Methodology; Writing—original draft; Writing—review and editing. **Premlata Kumari**: Resources; Investigation; Methodology; Writing—original draft; Writing—review and editing. **Bhabasha Gyanadeep Utakalaja**: Resources; Investigation; Methodology; Writing—original draft; Writing—review and editing. **Shraddheya Kumar Patel**: Investigation; Methodology; Writing—original draft; Writing—review and editing. **Narottam Acharya**: Conceptualization; Resources; Formal analysis; Supervision; Funding acquisition; Validation; Visualization; Writing—original draft; Project administration; Writing—review and editing.

Source data underlying figure panels in this paper may have individual authorship assigned. Where available, figure panel/source data authorship is listed in the following database record: biostudies:S-SCDT-10_1038-S44321-024-00080-8.

## Disclosure and competing interests statement

NA and SKP are listed as inventors in a related patent application. The remaining authors declare no competing interests.

# Expanded View Figures

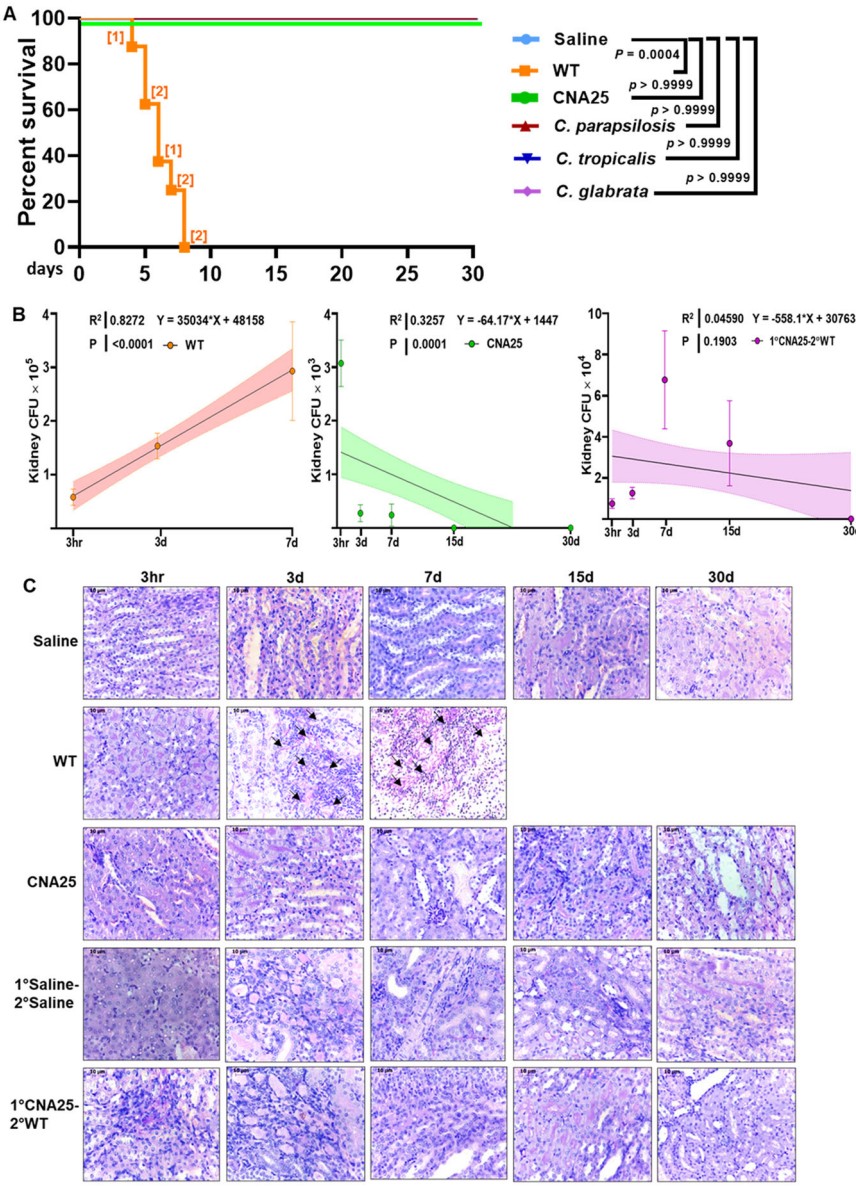

**Figure EV1. Virulence of NAC-species and a direct link between disease severity with fungal burden.**

(A) A Kaplan–Meier survival analysis of BALB/c mice (n = 8) upon intravenous administration of $5 \times 10^5$ CFU/mice of WT *C. albicans*, CNA25, *C. parapsilosis*, *C. tropicalis*, *C. glabrata*, and saline control (100 μl 1× PBS) for 30 days was conducted. Statistical significance among the comparing groups was determined using the log-rank (Mantel–Cox) test and the *P* values are listed as per the comparisons. (B) A simple linear regression analysis of kidney fungal burden in WT, CNA25, and 1°CNA25-2°WT with time (3 h, 3d, 7d, 15d, and 30d) and respective $R^2$, Equation and p values are given. (C) A time kinetic representative kidney PAS staining images of each group (Saline, WT, CNA25, 1°Saline-2°Saline, and 1°CNA25-2°WT) depicting fungal load post-inoculation are shown. Images were acquired in a Leica- ICC50 microscope at ×40 with a 10 μm scale bar. The black arrows indicate hyphal fungal cells.

## A. Circulating Myeloid cells

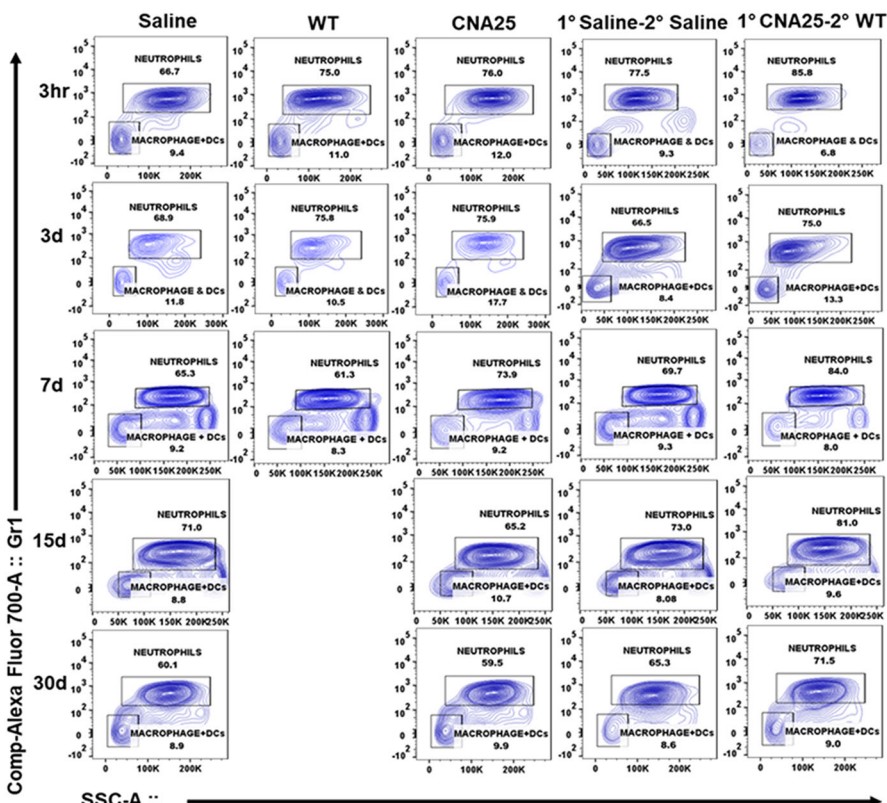

## B. Circulating Lymphoid cells

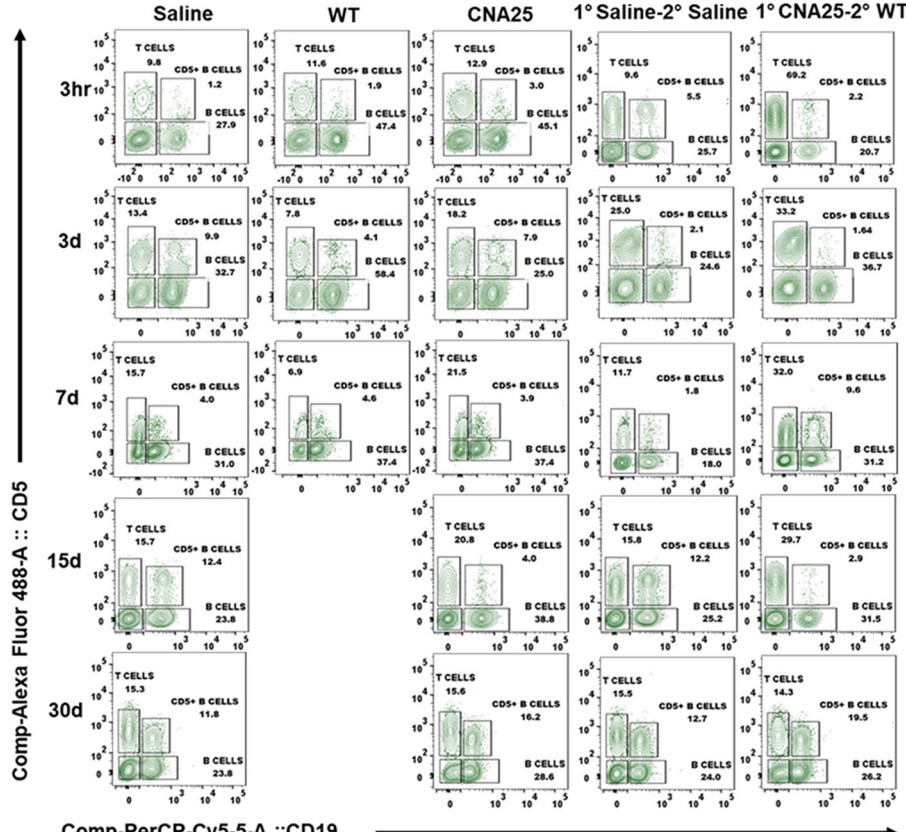

**Figure EV2.   Circulating myleoid and lymphoid immune cells profiling in various fungal-challenged mice groups.**

(A) Representative bivariate contour plots of all time points (3 h, 3d, 7d, 15d, and 30d) having Alexa Fluor 700 conjugated Gr-1 Ly6G/Ly6C on y axis and Side scatter on x axis for the analysis of compartmental distribution of circulating myeloid-derived Macrophage + DCs and neutrophils cells using Flowjo v8.0.2 Software are provided. (B) Similar bivariate contour plots of all time points having Alexa Fluor 488 conjugated CD5 on y axis and PerCP-Cy5.5 conjugated CD19 on x axis for the analysis of the compartmental distribution of circulating lymphoid-derived B, B1, and T cells using Flowjo v8.0.2 Software are given.

## A. Splenic Myeloid cells

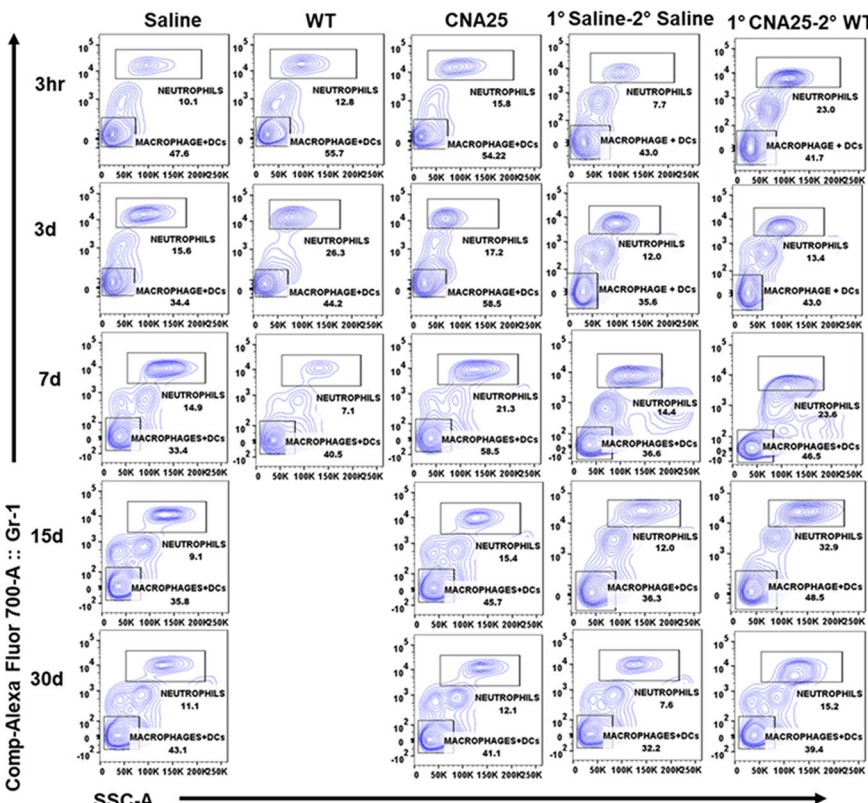

## B. Splenic Lymphoid cells

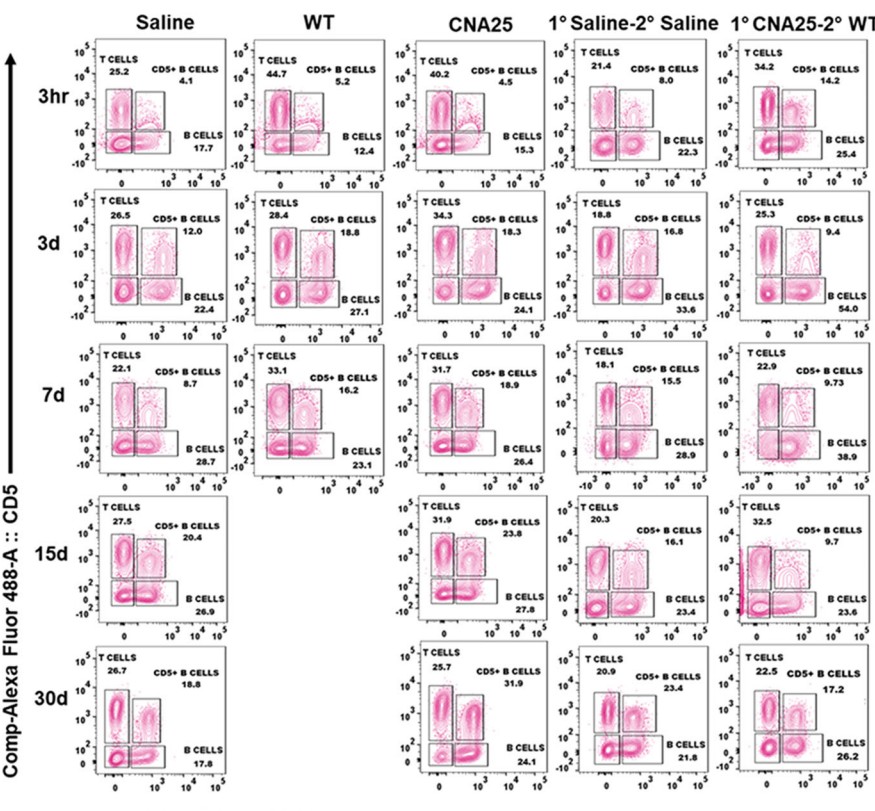

◄ **Figure EV3.** **Tissue-resident myleoid and lymphoid immune cells profiling in various fungal-challenged mice groups.**

(A) Representative time kinetic bivariate contour plots having Alexa Fluor 700 conjugated Gr-1 Ly6G/Ly6C on the y axis and Side scatter on the x axis for the analysis of compartmental distribution of splenic myeloid-derived Macrophage + DCs and neutrophils cells using Flowjo v8.0.2 Software are provided. (B) Similar bivariate contour plots of each time point having Alexa Fluor 488 conjugated CD5 on the y axis and PerCP-Cy5.5 conjugated CD19 on the x axis for the analysis of compartmental distribution of splenic lymphoid-derived B, B1, and T cells using Flowjo v8.0.2 Software are shown.

## Splenic CD4⁺ T cells subpopulation

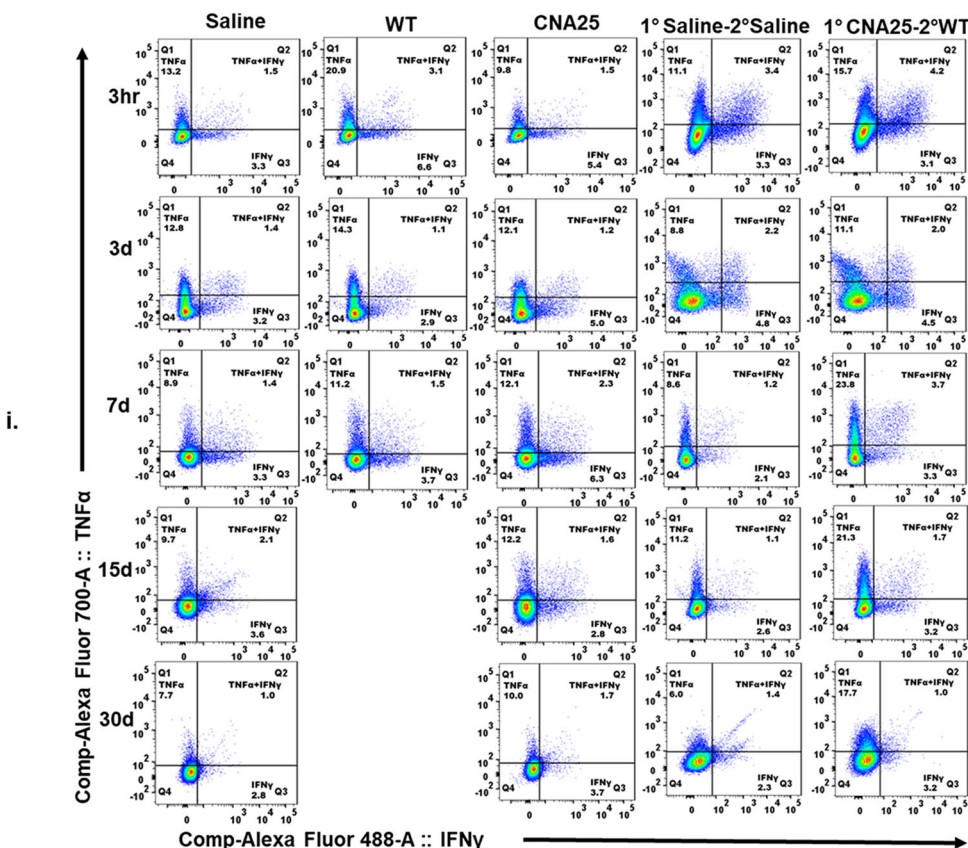

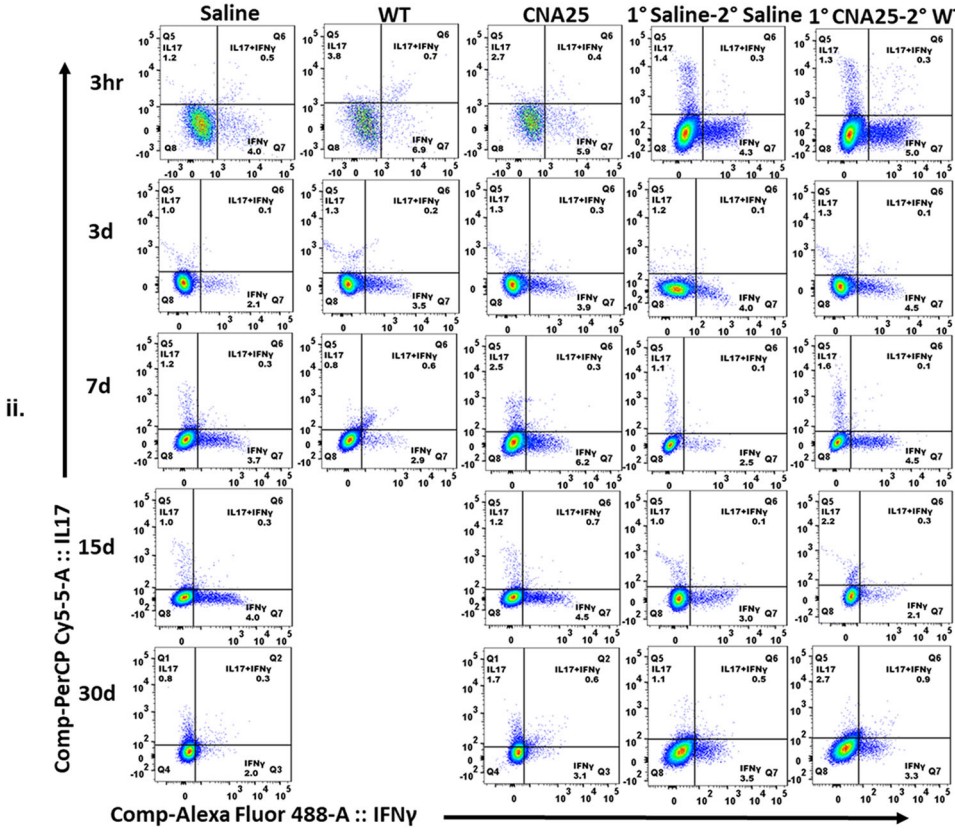

**Figure EV4.   CD4+ T cells profiling based on cytokine in various fungal-challenged mice groups.**

(i) Representative bivariate pseudo plots of all time points having Alexa Fluor 700 conjugated TNFα on the y axis and Alexa fluor 488 conjugated IFNγ on the x axis for the analysis of compartmental distribution of splenic CD4+ T cells subpopulation (IFNγ+ CD4+ T cells, TNFα+ CD4+ T cells, and IFNγ+TNFα+ CD4+ T cells) using Flowjo v8.0.2 Software are given. (ii) Representative bivariate pseudo plots of all time points having PerCP-Cy5.5 conjugated IL-17 on the y axis and Alexa fluor 488 conjugated IFNγ on the x axis for the analysis of the compartmental distribution of splenic CD4+T cells subpopulation (IL-17+ CD4+ T cells, IFNγ+ CD4+ T cells, and IL-17+IFNγ + CD4+ T cells) using Flowjo v8.0.2 Software are shown.

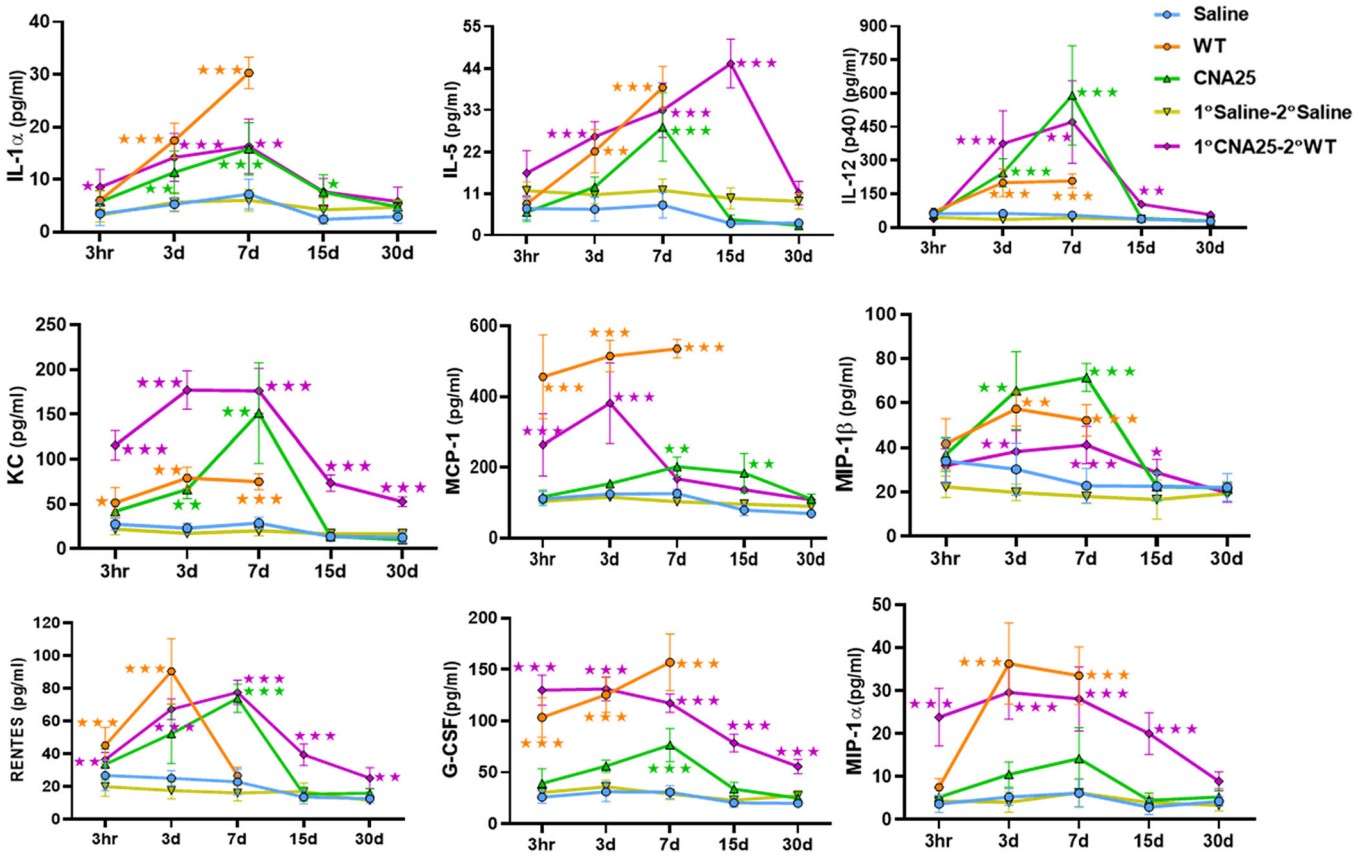

**Figure EV5.** **Serum cytokine profile of infected mice.**

The line graphs depicting the concentration of various cytokines and chemokines such as IL-1α, IL-5, IL-12(p40), KC, MCP-1, MIP-1β, MIP-1α, G-CSF, and RENTES in picogram/ml of each mouse of infected groups (WT—Orange, CNA25—green, Saline—sky blue, 1°Saline-2°Saline—greenish yellow, and 1°CNA25-2°WT—purple) at mentioned time points. The lines join the mean± SEM of eight mice data for each group and time point. Data are the representative of two separate experiments and were analyzed using the two-way ANOVA test (Tukey's multiple comparisons test). *P ≤ 0.05, **P ≤ 0.01, ***P ≤ 0.001, and no star symbols suggest nonsignificant. 1° is primary and 2° denotes re-challenge.

