## [Peer Review File · EMBO Molecular Medicine]

Immunogenicity and efficacy of CNA25 as a potential whole-cell vaccine against systemic candidiasis

Satya Sahu, Abinash Dutta, Doureradjou Peroumal, Premlata Kumari, Bhabasha Utakalaja, Shraddheya Patel, and Narottam Acharya

Corresponding author: Narottam Acharya (narottam_acharya@ils.res.in)

Review Timeline:

Submission Date:	11th Dec 23
Editorial Decision:	18th Jan 24
Revision Received:	11th Apr 24
Editorial Decision:	30th Apr 24
Revision Received:	10th May 24
Accepted:	10th May 24

Editor: Zeljko Durdevic

Transaction Report:

18th Jan 2024

Dear Dr. Acharya,

Thank you for the submission of your manuscript to EMBO Molecular Medicine, and please accept my apologies for the delay in getting back to you. We have now received feedback from two of the three reviewers who agreed to evaluate your manuscript. As the referee #2 will unfortunately not be able to return his/her report in a timely manner, and given that both reviewers provide very similar recommendations, we prefer to make a decision now in order to avoid further delay in the process.

As you will see from their reports pasted below, both referees recognize potential interest of the study but also raise serious concerns that should be addressed in a major revision. Particular attention should be given to providing more mechanistic insight and performing adequate statistical analyses. Should referee #2 provide a report, we will send it to you, with the understanding that we will not ask for an additional revision. If you would like to discuss further the points raised by the referees, I am available to do so via email or video. Let me know if you are interested in this option.

We would welcome the submission of a revised version within three months for further consideration. Please let us know if you require longer to complete the revision.

I look forward to receiving your revised manuscript.

Yours sincerely,

Zeljko Durdevic

We require:

- 1) A .docx formatted version of the manuscript text (including legends for main figures, EV figures and tables). Please make sure that the changes are highlighted to be clearly visible.
- 2) Individual production quality figure files as .eps, .tif, .jpg (one file per figure). For guidance, download the 'Figure Guide PDF': (<https://www.embopress.org/page/journal/17574684/authorguide#figureformat>).
- 3) A .docx formatted letter INCLUDING the reviewers' reports and your detailed point-by-point responses to their comments. As part of the EMBO Press transparent editorial process, the point-by-point response is part of the Review Process File (RPF), which will be published alongside your paper.
- 4) A complete author checklist, which you can download from our author guidelines (<https://www.embopress.org/page/journal/17574684/authorguide#submissionofrevisions>). Please insert information in the checklist that is also reflected in the manuscript. The completed author checklist will also be part of the RPF.

6) It is mandatory to include a 'Data Availability' section after the Materials and Methods. Before submitting your revision, primary datasets produced in this study need to be deposited in an appropriate public database, and the accession numbers and database listed under 'Data Availability'. Please remember to provide a reviewer password if the datasets are not yet public (see <https://www.embopress.org/page/journal/17574684/authorguide#dataavailability>).

13) Author contributions: You will be asked to provide CRediT (Contributor Role Taxonomy) terms in the submission system. These replace a narrative author contribution section in the manuscript.

14) A Conflict of Interest statement should be provided in the main text.

15) Every published paper now includes a 'Synopsis' to further enhance discoverability. Synopses are displayed on the journal webpage and are freely accessible to all readers. They include a short stand first (maximum of 300 characters, including space) as well as 2-5 one-sentence bullet points that summarize the paper. Please write the bullet points to summarize the key NEW findings. They should be designed to be complementary to the abstract - i.e. not repeat the same text. We encourage inclusion of key acronyms and quantitative information (maximum of 30 words / bullet point). Please use the passive voice. Please attach these in a separate file or send them by email, we will incorporate them accordingly.

Please also suggest a striking image or visual abstract to illustrate your article as a PNG file 550 px wide x 300-800 px high.

**** Reviewer's comments ****

Referee #1 (Comments on Novelty/Model System for Author):

In this work, the authors report the use of an avirulent *Candida albicans* strain as a vaccine strain. As such, their results on the strain's utility are solid. However, the study fails to unravel the mechanistic details of how this vaccine strain induces protection. Specifically, the study has not revealed the cellular and molecular pathways that are exploited by the vaccine strain. For example, it is reported that trained immunity induced in the myeloid phagocytes can promote protection upon rechallenge. Similarly, the direct examination of distinct immune subsets for their roles in promoting protection has not been investigated. These investigations can be implemented using mice with specific depletion of distinct cell subsets (for example, via antibody-mediated or clodronate-based approaches). Similarly, the authors have characterized the different molecules that are differentially regulated upon vaccination and rechallenge, but none have been mechanistically investigated for their relevance. If cellular and molecular mechanistic insights are provided, this study can be impactful.

Referee #1 (Remarks for Author):

In this manuscript by Sahu and Dutta et al, the authors report that a DNA polymerase mutant *Candida albicans* is a good vaccine candidate and can promote protection against murine systemic candidiasis due to *albicans* and non-*albicans* *Candida* species. Overall, the authors report a promising vaccine candidate that can benefit the critical unmet need posed by systemic candidiasis. The authors report that the vaccine strain can protect the mice from lethal rechallenge infection from *albicans* and non-*albicans* *Candida* species. Such protection is associated with systemic changes in leukocyte populations and circulating cytokines. Overall, the work is interesting and addresses a critical unmet need. However, there are certain points that diminish this reviewer's enthusiasm for the work in its current form. These are highlighted below:

1. The results on the strain's utility are solid. However, the study fails to unravel the mechanistic details of how this vaccine strain induces protection. Specifically, the study has not revealed the cellular and molecular pathways that are exploited by the vaccine strain. For example, it is known that trained immunity induced in the myeloid phagocytes can promote protection upon rechallenge (<https://pubmed.ncbi.nlm.nih.gov/34663098/>; <https://pubmed.ncbi.nlm.nih.gov/29339423/>; <https://pubmed.ncbi.nlm.nih.gov/25258083/>; <https://pubmed.ncbi.nlm.nih.gov/31085710/>). Is the vaccine strain promoting protection via trained immunity? Is the vaccine-induced protection dependent upon myeloid mononuclear phagocytes and/or granulocytes and/or lymphocytes? Their experiments with SCID and cyclophosphamide provide some hints but the above questions have not been investigated. These investigations can be implemented using mice with specific depletion of distinct cell subsets (for example, via antibody-mediated or clodronate-based approaches).

2. The authors have characterized the different cytokine/chemokine molecules that are differentially regulated upon vaccination and rechallenge, but none have been mechanistically investigated for their relevance. For example, the PRRs are differentially regulated in their findings; is the protection dependent upon DECTIN-1 and/or TLR2 or other PRRs? Are IFN γ or IL17a required or dispensable for vaccine-induced protection?

3. Prior studies (specified in my comment 1 above), have examined other *Candida* strains which promote protection upon systemic rechallenge. Bone-marrow infiltration by low virulence strains has been attributed with enhanced protection (<https://pubmed.ncbi.nlm.nih.gov/31085710/>). Does a similar mechanism explain the enhanced protection observed in the current setting?

Minor points:

1. Figure legends should specify the exact statistical test for each subpanel. Many figures currently do not have this information for all the included subpanels.
2. The survival curves depict numbers for each group on the curves. This reviewer was uncertain about what these numbers indicate. This should be specified.

Referee #3 (Comments on Novelty/Model System for Author):

The techniques used in this manuscript are solid, and the authors report a large amount of data. However, there are multiple problems with the statistical analysis of these data.

Referee #3 (Remarks for Author):

In the manuscript "Immunogenicity and efficacy of CNA25 as a potential whole cell vaccine against systemic candidiasis", Sahu et al. report the results of extensive experiments their group made to test if a *Candida albicans* mutant could work as a vaccine in mice. The authors show that the mutant is avirulent and that it protects mice against infection not only with *C. albicans*, but also with other non-*albicans* *Candida* species. The authors should be commended for their effort, as the manuscript is the product of a significant amount of work. The subject they address is of medical importance, and fits well within the scope of the journal.

There are, however, important issues with the manuscript text and with the analysis and reporting of data. I have organized them below.

Major issues:

- 1) The statistical analyses performed by the authors have multiple problems. I do not believe this precludes publication, but the authors need to do an extensive new analysis of most of their data.
- 2) The representation of the data collected by the authors on figures makes it hard to interpret their results. I believe they should revise several manuscript figures as well.

To organize the specific problems I see with statistical analyses and figures, I have organized them below by figures and/or Results sub-sections.

Figure 1, pages 4 and 5:

- a) The experiments with mice shown in Figures 1A, 1B 1C and 1D do not have any kind of inferential statistics analysis, such as p-values or confidence intervals. No inferential statistics analyses for these experiments are also described in the results section text or in the materials and methods. However, the authors make several inferences from these data, and these conclusions are central to the manuscript. It is crucial to properly analyze these data using adequate statistical tools.
- b) Figure 1, panel B-i: The y-axis with a linear scale and with two breaks can be misleading. This graph should be presented with logarithmic scale and without the breaks on the y-axis.
- c) Page 5, lines 12-13: "All the protected mice that survived beyond 30 days in all our experiments were allowed to survive till their natural death unless mentioned." These data should be shown, then. It is important to evaluate whether vaccination with CNA25 prevented death altogether or if it simply delayed deaths beyond day 30 post-infection.

Figure 2, pages 5-6:

- d) Experiments shown on Figure 2 also do not have any kind of inferential statistics analyses, which I believe are crucial as explained above.
- e) Figure 2, panels B and C: The y-axes in these figures should also be presented with a logarithmic scale, without the breaks. Additionally, the x-axis in these figures represents a continuous variable (days); it is thus better to use line graphs instead of bar graphs which are more appropriate for categorical data. Moreover, the authors could perform statistical analyses that are appropriate for continuous variables such as regression.

Figure 3, pages 6-7:

f) Data on red blood cells and platelets do not show differences that are biologically significant, in my opinion. This makes sense, because clinically I would not expect either systemic candidiasis or vaccination to result in drastic changes to these cells. I would thus suggest that these data be omitted.

g) Each of these experiments shown on Figure 3 have one dependent continuous variable (cell count) and three independent variables (time after infection, vaccination status and what was used for the challenge (saline versus *C. albicans*); one of these independent variables is continuous. The authors use two-way ANOVA to analyze this dataset, which is used to assess the effect of two independent categorical variables on a continuous categorical variable. Thus, I believe the tool chosen for the statistical analysis is not optimal. Perhaps a generalized linear model regression would be more appropriate in this case.

h) Figure 3 has many panels, each one with a lot of bars. This representation makes it very time-consuming and confusing to interpret the results. There are many other tools that could be used to represent these data on a way that is clearer and more concise.

Figure 4, pages 7-8:

i) The gating strategy for this phagocytosis assay is not shown. Were the cells gated on forward versus side scatter graphs to remove debris and large aggregates? Were they then gated to remove doublets?

j) Were single-color controls used? Or fluorescence-minus-one controls? These would be important in validating the position of the gates the authors have drawn, which is crucial given that the populations of CFSE-positive and CFSE-negative events are not separated.

k) Figure 4B lacks a legend explaining what each color is.

l) Figure 4C. This experiment measures the proportion of macrophages that are PI-positive. The inferential statistics analysis tool the authors chose (ANOVA) is for continuous, not proportion dependent variables. They should use instead Chi square or Fisher exact test instead.

Figure 5, page 8:

m) The entire section on "differential expression profile of pattern recognition receptors (PRRs) in fungal infected mice" is problematic, in my opinion. Measuring mRNA levels does not necessarily correlate with activation of a certain pathway. For example, mRNA levels could be increased without an increase in protein levels, due to RNA transport or stability regulation. Most importantly, however, cells use post-translational modification to control several of the pathways controlled by the genes the authors measured. Additionally, increases in the expression of these genes could just reflect the kidney infiltration by leukocytes the authors measured in experiments shown earlier in the manuscript, as leukocytes probably express more of these receptors than kidney cells. I thus believe this entire section should be removed from the paper, or at least re-written.

Figure 6, page 9:

n) The authors chose to use CD5 as a T-cell marker, instead of the most common (and more sensitive) CD3. I do not believe this is a problem, but there is actually an interesting finding on their data that should be explored. CD5 is expressed on B1 cells, a subpopulation of B cells with interesting characteristics. Several plots on Figures 6, S2 and S3 show a population of CD19-positive, CD5-positive population. I wonder if these are B1 cells. This could be interesting because some figures show a clear difference in this population.

o) Figure 6 presently occupies three pages. There are certainly ways to present these data more concisely and clearly. Several plots could also be put on the supplement.

p) As explained above, a line graph might be a better representation for continuous variables such as days instead of bar graphs.

Figure 7, page 10:

q) Again the authors use ANOVA to analyse proportion variables. As explained above, Chi square or Fisher's exact test might be better.

r) This figure also occupies multiple pages, so it should be optimized.

Figure 8, pages 10-11:

s) Several issues mentioned above are repeated in this figure, such as it occupying three pages, having multiple bar graphs, using linear and broken y-axes and representing a continuous variable on a bar graph.

Figure 9, pages 12-13:

t) This figure occupies two pages, so it should be optimized.

u) Multiple experiments are shown without inferential statistics analyses. It is crucial that proper analyses be made.

Other major issues:

3) Page 10, lines 13 - 15: "Our result indicated that the early activation of CD4+ and CD8+ T cells is required for fungal clearance, whereas CD4+ T cells are most likely involved in protective immune memory response": The data presented do not allow this type of conclusion. Cell counts are observational experiments, so at best the authors could speculate that T cells could be required. An affirmation such as this would require more complex experiments, such as gain-of-function or loss-of-function assays. The authors should either change their conclusions on the manuscript text or perform the experiments which are necessary to reach such a conclusion.

4) After the magnetic isolation of T cells for subpopulation studies, did the authors confirm how pure were the isolated cells? Given that the results shown on figure 7 show percentages of cytokine-positive cells ranging from 0.1 to 6.2, even small populations of contaminating cells (e.g. TNF-alpha producing monocytes) could skew the results.

Minor issues>

5) Line 3: "About billions get infected". This phrase needs to be corrected.

6) Page 4, line 22: Perhaps the results for this experiment could be added to the supplement, instead of just reporting them as "data not shown."

7) Page 4, lines 19 - 22. "With the same inoculum size (5 x 10⁵ CFU) and in terms of fatality rate within 30 days of infection period, while *C. albicans* killed the animals within 10 days of infection, the non-*albicans* strains did not cause any animal death (data not shown)." The authors start by mentioning mortality in 30 days, only to immediately express results in terms of how long it took for all animals to die. The text here would be clearer if the authors choose a single measure to compare the outcomes of infection with different *Candida* species.

8) Page 4, line 23: It is not clear what the authors mean by "delayed death rate" here. It is better to state actual experimental results, such as median time to death or 30-day survival rates.

9) The researchers use both "mortality" (page 3, line 13; page 6, 30, page 13, line 18) and "fatality" (page 2, line 2; page 4, line 20) interchangeably throughout the paper. Despite their similarities, in epidemiology they have different meanings. It is important that the authors clarify what they mean and use the most appropriate expression consistently.

10) "FACS" (page 7, line 23) is a trademark. I would suggest using flow cytometry instead.

11) Page 11, line 17: The authors add IL-2 to a group of anti-inflammatory cytokines. However, this cytokine is an important T cell activator, produced in large quantities by pro-inflammatory Th1 cells. This should be corrected.

12) Page 13, line 11: "Despite fungi being as deadly as viruses and bacteria". This phrase is not accurate. Invasive mycoses do kill many people, but there are multiple examples of viral and bacterial diseases with higher mortality and case-fatality rates in comparison with fungal diseases.

13) Page 17, line 29 (and other instances): What do the authors mean by "shorted"?

14) Page 18, lines 6-7. The authors should describe the equipment used for fluorescence imaging. Some information is given on the legend for figure 4, but more details should be given on the Materials and Methods section.

15) The discussion section is very long. Moreover, parts of the Results section read more like introduction or discussion texts. The authors should revise the text to make it more concise.

EMBO Molecular Medicine

April 11th, 2024

Dear Editor,

Thank you very much for the review of our manuscript "Immunogenicity and efficacy of CNA25 as a potential whole-cell vaccine against systemic candidiasis" (EMM-2023-19129). We would like to thank the Editor and reviewers for their valuable time, careful reading, and constructive comments. The comments have been highly valuable and useful for improving the quality of our study, as well as important in guiding the direction of our present and future research. In the revised manuscript, we have incorporated the necessary changes including additional experimental data as suggested. Please find our detailed point-by-point response to the reviewer's comments and the changes we have made in the manuscript as follows.

Referee #1 (Comments on Novelty/Model System for Author):

In this work, the authors report the use of an avirulent *Candida albicans* strain as a vaccine strain. As such, their results on the strain's utility are solid. However, the study fails to unravel the mechanistic details of how this vaccine strain induces protection. Specifically, the study has not revealed the cellular and molecular pathways that are exploited by the vaccine strain. For example, it is reported that trained immunity induced in the myeloid phagocytes can promote protection upon rechallenge. Similarly, the direct examination of distinct immune subsets for their roles in promoting protection has not been investigated. These investigations can be implemented using mice with specific depletion of distinct cell subsets (for example, via antibody-mediated or clodronate-based approaches). Similarly, the authors have characterized the different molecules that are differentially regulated upon vaccination and rechallenge, but none have been mechanistically investigated for their relevance. If cellular and molecular mechanistic insights are provided, this study can be impactful.

Reply: Thank you so much for appreciating our study and the utility of our vaccine strain. As suggested, the cellular and molecular mechanisms of protective immunity in CNA25 immunized mice have now been provided in the revised manuscript and the details can be found below.

Referee #1 (Remarks for Author):

In this manuscript by Sahu and Dutta et al, the authors report that a DNA polymerase mutant *Candida albicans* is a good vaccine candidate and can promote protection against murine systemic candidiasis due to *albicans* and non-*albicans* *Candida* species. **Overall, the authors report a promising vaccine candidate that can benefit the critical unmet need posed by systemic candidiasis.** The authors report that the vaccine strain can protect the mice from lethal rechallenge infection from *albicans* and non-*albicans* *Candida* species. **Such protection is associated with systemic changes in leukocyte populations and circulating cytokines.** Overall, the work is interesting and addresses a critical

unmet need. However, there are certain points that diminish this reviewer's enthusiasm for the work in its current form. These are highlighted below:

1. The results on **the strain's utility are solid.** However, the study fails to unravel the mechanistic details of how this vaccine strain induces protection. Specifically, the study has not revealed the cellular and molecular pathways that are exploited by the vaccine strain. For example, it is known that trained immunity induced in the myeloid phagocytes can promote protection upon rechallenge (<https://pubmed.ncbi.nlm.nih.gov/34663098/>; <https://pubmed.ncbi.nlm.nih.gov/29339423/>; <https://pubmed.ncbi.nlm.nih.gov/25258083/>; <https://pubmed.ncbi.nlm.nih.gov/31085710/>). Is the vaccine strain promoting protection via trained immunity? Is the vaccine-induced protection dependent upon myeloid mononuclear phagocytes and/or granulocytes and/or lymphocytes? Their experiments with SCID and cyclophosphamide provide some hints but the above questions have not been investigated. These investigations can be implemented using mice with specific depletion of distinct cell subsets (for example, via antibody-mediated or clodronate-based approaches).

Reply: As appreciated by the respected reviewer, we have already provided some evidence of immune cell subtypes involved in fungal clearance by using SCID and CPM-induced immunosuppressive models. To strengthen it further, we used specific antibody-mediated and clodronate-based cellular depletion approaches to determine the significance of various immune cells induced in vaccinated mice and their involvement in protective immune responses. Using a standardized protocol, anti-Gr-1 antibody, anti-CD4 antibody, anti-CD8 antibody, and clodronate phosphate were administered to the vaccinated mice intraperitoneally to deplete PMNLs, CD4⁺ T cells, CD8⁺ T cells, and macrophages, respectively, and further, they were re-challenged with WT *C. albicans* cells and survivability was monitored (**Please see Fig. 7B and 9B**). Our results suggested the involvement of both adaptive and trained immunity in CNA25 vaccinated mice protecting lethal re-challenge.

2. The authors have characterized the different cytokine/chemokine molecules that are differentially regulated upon vaccination and rechallenge, but none have been mechanistically investigated for their relevance. For example, the PRRs are differentially regulated in their findings; is the protection dependent upon DECTIN-1 and/or TLR2 or other PRRs? Are IFN γ or IL17a required or dispensable for vaccine-induced protection?

Reply- Similarly, as above anti-Dectin-1 antibody, anti-TLR2 antibody, anti-IL17A antibody, anti-IFN γ antibody, and anti-TNF α antibody mediated blocking of the specific molecules in CNA25 vaccinated mice was carried out and their survivability was monitored upon WT re-challenge (**Please**

see Fig. 5B and 8B). Blocking of these molecules suppressed resistance to re-infection in CNA25 vaccinated mice except in the anti-TLR2 antibody blocked group suggesting the involvement of these molecules in protective immunity.

3. Prior studies (specified in my comment 1 above), have examined other *Candida* strains which promote protection upon systemic rechallenge. Bone-marrow infiltration by low virulence strains has been attributed with enhanced protection (<https://pubmed.ncbi.nlm.nih.gov/31085710/>). Does a similar mechanism explain the enhanced protection observed in the current setting?

Reply- Although the bone marrow infiltration of less-virulent strains is associated with protection, in our study we detected infiltration of both WT and CNA25 (virulent and avirulent strains) fungal cells in the femoral bone marrow in equal efficiency in the pre-challenged mice post- 24 and 48 hrs of IV as well as IP inoculation. Therefore, we have not provided this data in the revised manuscript.

Minor points:

Figure legends should specify the exact statistical test for each subpanel. Many figures currently do not have this information for all the included subpanels.

Reply- Thank you so much. The necessary information has been provided.

2. The survival curves depict numbers for each group on the curves. This reviewer was uncertain about what these numbers indicate. This should be specified.

Reply- The information has now been provided in the figure legend. The numerical values in the survival graph indicate the number of mice that succumbed to infection on a given day.

Referee #3 (Comments on Novelty/Model System for Author):

The techniques used in this manuscript are solid, and the authors report a large amount of data. However, there are multiple problems with the statistical analysis of these data.

Reply- Thank you very much for appreciating our work and in the revised manuscript specific statistical analysis for each experiment was conducted as recommended and has now been provided in the revised manuscript.

Referee #3 (Remarks for Author):

In the manuscript "Immunogenicity and efficacy of CNA25 as a potential whole cell vaccine against systemic candidiasis", Sahu et al. report the results of extensive experiments their group made to test if a *Candida albicans* mutant could work as a vaccine in mice. The authors show that the mutant is avirulent and that it protects mice against infection not only with *C. albicans*, but also with other non-*albicans Candida* species. **The authors should be commended for their effort, as the manuscript is the product of a significant amount of work. The subject they address is of medical importance, and fits well within the scope of the journal.**

Reply- Thank you so much for appreciating our effort to identify and demonstrate a whole-cell vaccine strain against fungal infections.

There are, however, important issues with the manuscript text and with the analysis and reporting of data. I have organized them below.

Major issues:

- 1) The statistical analyses performed by the authors have multiple problems. I do not believe this precludes publication, but the authors need to do an extensive new analysis of most of their data.
- 2) The representation of the data collected by the authors on figures makes it hard to interpret their results. I believe they should revise several manuscript figures as well. To organize the specific problems I see with statistical analyses and figures, I have organized them below by figures and/or Results sub-sections.

Reply- Thank you so much for helping us to re-organise our figures with proper statistical analysis.

Figure 1, pages 4 and 5:

- a) The experiments with mice shown in Figures 1A, 1B, 1C and 1D do not have any kind of inferential statistics analysis, such as p-values or confidence intervals. No inferential statistics analyses for these experiments are also described in the results section text or in the materials and methods. However, the authors make several inferences from these data, and these conclusions are central to the manuscript. It is crucial to properly analyze these data using adequate statistical tools.

Reply- Thank you so much for pointing this out. All the mice survival analyses have now been statically analysed by using the log-rank (Mantel-Cox) test. The actual p values are listed beside the corresponding comparison groups.

b) Figure 1, panel B-i: The y-axis with a linear scale and with two breaks can be misleading. This graph should be presented with logarithmic scale and without the breaks on the y-axis.

Reply- As suggested all the CFU analyses were now re-analysed and presented in the Log_{10} scale. The breaks in the y-axis were removed.

c) Page 5, lines 12-13: "All the protected mice that survived beyond 30 days in all our experiments were allowed to survive till their natural death unless mentioned." These data should be shown, then. It is important to evaluate whether vaccination with CNA25 prevented death altogether or if it simply delayed deaths beyond day 30 post-infection.

Reply- As has been consistently shown in this study, 100% of the mice succumb to the lethal challenge of WT fungal infection within 15 days of inoculation. 30 days of survival period is a standard way of representing a survival curve. In fact, we have followed the longevity of these surviving mice for 6 months and after this observation period, they were handed over to the animal house for proper clearance as per the institutional guidelines (This has now been mentioned in the text). Also, most of our analyses suggested no or hardly any fungal load in the vital organs of survived mice beyond 15 days of inoculation. Therefore, we believe these mice will have their natural death independent of fungal infection.

Figure 2, pages 5-6:

d) Experiments shown on Figure 2 also do not have any kind of inferential statistics analyses, which I believe are crucial as explained above.

Reply- The CFU analyses in Fig. 2B were re-analysed using Mann-Whitney U test and presented in the Log_{10} scale. Thank you for pointing this out.

e) Figure 2, panels B and C: The y-axes in these figures should also be presented with a logarithmic scale, without the breaks. Additionally, the x-axis in these figures represents a continuous variable (days); it is thus better to use line graphs instead of bar graphs which are more appropriate for categorical data. Moreover, the authors could perform statistical analyses that are appropriate for continuous variables such as regression.

Reply- The CFU analyses in Fig. 2B were now re-analysed using the Mann-Whitney U test and presented in the Log_{10} scale. Fig. 2C is now moved as a supplementary figure as Fig. EV1C and regression analysis of fungal load in kidney Vs days of infection in a particular mice group is provided in Fig. EV1B.

Figure 3, pages 6-7:

f) Data on red blood cells and platelets do not show differences that are biologically significant, in my opinion. This makes sense, because clinically I would not expect either systemic candidiasis or vaccination to result in drastic changes to these cells. I would thus suggest that these data be omitted.

Reply- We agree with you and RBC levels are not changing significantly, thus it is now removed. However, the platelets are changing and the regression analysis also clearly suggested certain trends in different mice groups, therefore it was retained in Fig.3.

g) Each of these experiments shown on Figure 3 have one dependent continuous variable (cell count) and three independent variables (time after infection, vaccination status and what was used for the challenge (saline versus *C. albicans*); one of these independent variables is continuous. The authors use two-way ANOVA to analyze this dataset, which is used to assess the effect of two independent categorical variables on a continuous categorical variable. Thus, I believe the tool chosen for the statistical analysis is not optimal. Perhaps a generalized linear model regression would be more appropriate in this case.

Reply- Thank you for the suggestion, we have now added regression analyses for this set of data besides the line graphs.

h) Figure 3 has many panels, each one with a lot of bars. This representation makes it very time-consuming and confusing to interpret the results. There are many other tools that could be used to represent these data on a way that is clearer and more concise.

Reply- Bar graphs have been replaced with line graphs for better comparison.

Figure 4, pages 7-8:

i) The gating strategy for this phagocytosis assay is not shown. Were the cells gated on forward versus side scatter graphs to remove debris and large aggregates? Were they then gated to remove doublets?

Reply- A detailed getting strategy of fungal-macrophage interaction has now been given **in Appendix Fig. S1A**. First, we removed the debris and then the doublets were removed. Only the singlets were considered for further analyses.

j) Were single-color controls used? Or fluorescence-minus-one controls? These would be important in validating the position of the gates the authors have drawn, which is crucial given that the populations of CFSE-positive and CFSE-negative events are not separated.

Reply- We completely agree with you. Various cell segregations are now given in Appendix Fig. S1A. In various control experiments, one can see no cell population in Q2 of the FACS plot.

k) Figure 4B lacks a legend explaining what each color is.

Reply- Thank you, it has now been added. We have followed the same colour pattern for each strain/mouse group in all the figures.

l) Figure 4C. This experiment measures the proportion of macrophages that are PI-positive. The inferential statistics analysis tool the authors chose (ANOVA) is for continuous, not proportion dependent variables. They should use instead Chi square or Fisher exact test instead.

Reply- As suggested the Chi square test is used for Fig. 4C analyses.

Figure 5, page 8:

m) The entire section on "differential expression profile of pattern recognition receptors (PRRs) in fungal infected mice" is problematic, in my opinion. Measuring mRNA levels does not necessarily correlate with activation of a certain pathway. For example, mRNA levels could be increased without an increase in protein levels, due to RNA transport or stability regulation. Most importantly, however, cells use post-translational modification to control several of the pathways controlled by the genes the authors measured. Additionally, increases in the expression of these genes could just reflect the kidney infiltration by leukocytes the authors measured in experiments shown earlier in the manuscript, as leukocytes probably express more of these receptors than kidney cells. I thus believe this entire section should be removed from the paper, or at least re-written.

Reply- Since the kidney is the primary organ where *C. albicans* colonise, we checked the expression of PRRs in the infected organs. Apart from the renal cells, as the respected reviewer mentioned, the high expression of PRRs in kidneys could also be due to infiltrated immune cells expressing PRRs. This portion of the text has been re-written. Since the PRR blocking experiment suggested a primary role of Dectin-1 but less likely TLR2 in protective immunity, the higher expression of these receptors and immune activation at various stages seem to be co-related (Fig. 5B, please see our response to Reviewer #1 Point 2).

Figure 6, page 9:

n) The authors chose to use CD5 as a T-cell marker, instead of the most common (and more sensitive) CD3. I do not believe this is a problem, but there is actually an interesting finding on their data that should be explored. CD5 is expressed on B1 cells, a subpopulation of B cells with interesting characteristics. Several plots on Figures 6, S2 and S3 show a population of CD19-positive, CD5-positive population. I wonder if these are B1 cells. This could be interesting because some figures show a clear difference in this population.

Reply- Thank you for carefully analysing our data, yes, these cells are CD5⁺ B cells. We have now estimated these populations and the data have been added to the revised manuscript (Fig. 6A v and Fig. 6B v).

o) Figure 6 presently occupies three pages. There are certainly ways to present these data more concisely and clearly. Several plots could also be put on the supplement.

Reply- We have now segregated this as circulating and tissue-resident immune cells (now it is Fig. 6 A and B). All the bi-variant plots were moved to supplementary sections EV2, EV3, EV4, and appendix figure S3.

p) As explained above, a line graph might be a better representation for continuous variables such as days instead of bar graphs.

Reply- As suggested bar-graphs were replaced with multi-coloured lines.

Figure 7, page 10:

q) Again the authors use ANOVA to analyse proportion variables. As explained above, Chi square or Fisher's exact test might be better.

Reply- We tried to use Chi square test for these analyses but it does not fit well probably as they are multi variants. Therefore, we retained two-way ANOVA for these analyses.

r) This figure also occupies multiple pages, so it should be optimized.

Reply- We have moved the bi-variant plots to the supplementary section and the bar-graphs were replaced with line graphs.

Figure 8, pages 10-11:

s) Several issues mentioned above are repeated in this figure, such as it occupying three pages, having multiple bar graphs, using linear and broken y-axes and representing a continuous variable on a bar graph.

Reply-We have now only retained line graphs of key cytokine molecules, rest were moved to the supplementary sections (EV5).

Figure 9, pages 12-13:

t) This figure occupies two pages, so it should be optimized.

Reply- Fig. 9 is modified as suggested.

u) Multiple experiments are shown without inferential statistics analyses. It is crucial that proper analyses be made.

Reply- Thank you so much for pointing this out. All the mice survival analyses have now been statically analysed by using the log-rank (Mantel-Cox) test. The actual *p* values are listed beside the corresponding comparison groups.

Other major issues:

3) Page 10, lines 13 - 15: "Our result indicated that the early activation of CD4+ and CD8+ T cells is required for fungal clearance, whereas CD4+ T cells are most likely involved in protective immune memory response": The data presented do not allow this type of conclusion. Cell counts are observational experiments, so at best the authors could speculate that T cells could be required. An affirmation such as this would require more complex experiments, such as gain-of-function or loss-of-function assays. The authors should either change their conclusions on the manuscript text or perform the experiments which are necessary to reach such a conclusion.

Reply- To confirm the possible role of CD4+ and CD8+ T cells in protective immunity, we carried out a cellular depletion assay by injecting anti-CD4, anti-CD8, and their isotypes intraperitoneally to the vaccinated mice, and their resistance to reinfection was monitored. This data has been added. (Fig. 7 B, Please see our response section of reviewer-1)

4) After the magnetic isolation of T cells for subpopulation studies, did the authors confirm how pure were the isolated cells? Given that the results shown on figure 7 show percentages of cytokine-positive cells ranging from 0.1 to 6.2, even small populations of contaminating cells (e.g. TNF-alpha producing monocytes) could skew the results.

Reply- Yes, we do check the purity of the T cells and a representative unstained and stained cells are given for reference. We usually get ~90% of CD4+ T cells and the rest are other T cells.

Minor issues>

5) Line 3: "About billions get infected". This phrase needs to be corrected.

Reply- The statement has been modified.

6) Page 4, line 22: Perhaps the results for this experiment could be added to the supplement, instead of just reporting them as "data not shown."

Reply- Now it has been added in the supplementary section EV1 A.

7) Page 4, lines 19 - 22. "With the same inoculum size (5×10^5 CFU) and in terms of fatality rate within 30 days of infection period, while *C. albicans* killed the animals within 10 days of infection, the non-albicans strains did not cause any animal death (data not shown)." The authors start by mentioning mortality in 30 days, only to immediately express results in terms of how long it took for all animals to die. The text here would be clearer if the authors choose a single measure to compare the outcomes of infection with different Candida species.

Reply- Yes, here only the mortality rate was determined. This portion is revised.

8) Page 4, line 23: It is not clear what the authors mean by "delayed death rate" here. It is better to state actual experimental results, such as median time to death or 30-day survival rates.

Reply- Thank you, we have now revised as suggested. A table for the median death time of each group of mice has now been provided (Table-EV1)

9) The researchers use both "mortality" (page 3, line 13; page 6, 30, page 13, line 18) and "fatality" (page 2, line 2; page 4, line 20) interchangeably throughout the paper. Despite their similarities, in epidemiology they have different meanings. It is important that the authors clarify what they mean and use the most appropriate expression consistently.

Reply- We have now retained mortality only.

10) "FACS" (page 7, line 23) is a trademark. I would suggest using flow cytometry instead.

Reply- Thank you, now it has been modified.

11) Page 11, line 17: The authors add IL-2 to a group of anti-inflammatory cytokines. However, this cytokine is an important T cell activator, produced in large quantities by pro-inflammatory Th1 cells. This should be corrected.

Reply- Thank you, it has been modified accordingly.

12) Page 13, line 11: "Despite fungi being as deadly as viruses and bacteria". This phrase is not accurate. Invasive mycoses do kill many people, but there are multiple examples of viral and bacterial diseases with higher mortality and case-fatality rates in comparison with fungal diseases.

Reply- Revised accordingly.

13) Page 17, line 29 (and other instances): What do the authors mean by "shorted"?

Reply- Revised the terms.

14) Page 18, lines 6-7. The authors should describe the equipment used for fluorescence imaging. Some information is given on the legend for figure 4, but more details should be given on the Materials and Methods section.

Reply- Necessary information has been added (EVOS imaging system).

15) The discussion section is very long. Moreover, parts of the Results section read more like introduction or discussion texts. The authors should revise the text to make it more concise.

Reply-We have now revised the text extensively to make the manuscript more concise and comprehensible with small paragraphs. Repetitive informations were removed.

Once again we thank the editors and reviewers for their valuable time spent reviewing our manuscript, we hope that the revised version will be acceptable for publication.

Sincerely,

Dr. Acharya

30th Apr 2024

Dear Dr. Acharya,

Thank you for the submission of your revised manuscript to EMBO Molecular Medicine. I am pleased to inform you that we will be able to accept your manuscript pending the following final amendments:

- 1) Please address all the referee #1 concerns.
- 2) We note that some panels are reused. Please check below and clarify. If the re-use is intentional, please specify this in the corresponding figure legend.
 - Figure 1B ii and Figur 9 A iii WT - Possible re-use of cell image.
 - Figure EV2B and Appendix Figure S1B - Possible re-use of FACS
 - Figure EV4 - Possible re-use within the figure.
 - Appendix Figure S1 C,D and Appendix Figure S3 A,B - Possible re-use of FACS.
- 3) In the main manuscript file, please do the following:
 - Please address all comments suggested by our data editors listed below:
 - o Figure legends:
 1. Please note that the figure legends 4(i-ii) is labeled as 4a-b in the manuscript. This needs to be rectified.
 2. Please indicate the statistical test used for data analysis in the legend of figure EV 1b.
 3. Please note that information related to n is missing in the legends of figures 1b (i); 3b; 4b; EV 5.
 4. Please note that the error bars are not defined in the legends of figures 1b (i); 3a-b; 4b; 9a (ii); EV 5.
 5. Please note that scale bar and its definition are missing for figure 4c (i).
 6. Please note that the black arrows are not defined in the legends of figures 2b (ii); 9c (iii). This needs to be rectified.
 - Reduce keywords to max. 5.
 - Remove data not shown on p. 7.
 - Remove list of abbreviations.
 - Please provide the antibody dilutions that were used for each antibody in the Table 1.
 - Rename "Competing interests" to "Disclosure Statement & Competing Interests". We updated our journal's competing interests policy in January 2022 and request authors to consider both actual and perceived competing interests. Please review the policy <https://www.embopress.org/competing-interests> and update your competing interests if necessary.
 - Author contributions: Please remove it from the manuscript and specify author contributions in our submission system. CRediT has replaced the traditional author contributions section because it offers a systematic machine-readable author contributions format that allows for more effective research assessment. You are encouraged to use the free text boxes beneath each contributing author's name to add specific details on the author's contribution. More information is available in our guide to authors: <https://www.embopress.org/page/journal/17574684/authorguide#authorshipguidelines>
 - Correct the reference citation in the reference list. Where there are more than 10 authors on a paper, 10 will be listed, followed by "et al.". Please check "Author Guidelines" for more information. <https://www.embopress.org/page/journal/17574684/authorguide#referencesformat>
 - Rename "Availability of data and materials" to "Data availability" and add the following text and information "This study includes no data deposited in external repositories. The source data of this paper are collected in the following database record: Accession number and URL to BioStudies.
- 4) Tables: Remove Table EV1 and Table EV2 from the manuscript text and upload them as two separate files.
- 5) Appendix: Remove keywords and move figure legends under the corresponding figure.
- 6) Funding: Please merge it with "Acknowledgments".
- 7) The Paper Explained: Please add it to the main manuscript file.
- 8) Synopsis:
 - Synopsis image: Please reformat the image to 550 px-wide x (250-400)-px high and upload it as a high-resolution jpeg file.
 - Please check your synopsis text and image before submission with your revised manuscript. Please be aware that in the proof stage minor corrections only are allowed (e.g., typos).
- 9) For more information: This space should be used to list relevant web links for further consultation by our readers. Could you identify some relevant ones and provide such information as well? Some examples are patient associations, relevant databases, OMIM/proteins/genes links, author's websites, etc...
- 10) As part of the EMBO Publications transparent editorial process initiative (see our Editorial at <http://embomolmed.embopress.org/content/2/9/329>), EMBO Molecular Medicine will publish online a Review Process File (RPF) to accompany accepted manuscripts. This file will be published in conjunction with your paper and will include the anonymous referee reports, your point-by-point response and all pertinent correspondence relating to the manuscript. Let us know whether you agree with the publication of the RPF and as here, if you want to remove or not any figures from it prior to publication. Please note that the Authors checklist will be published at the end of the RPF.
- 11) Please provide a point-by-point letter INCLUDING my comments as well as the reviewer's reports and your detailed responses (as Word file).

I look forward to reading a new revised version of your manuscript as soon as possible.

Yours sincerely,

Zeljko Durdevic

*** Instructions to submit your revised manuscript ***

- 1) a .docx formatted version of the manuscript text (including Figure legends and tables)
- 2) Separate figure files*
- 3) supplemental information as Expanded View and/or Appendix. Please carefully check the authors guidelines for formatting Expanded view and Appendix figures and tables at <https://www.embopress.org/page/journal/17574684/authorguide#expandedview>
- 4) a letter INCLUDING the reviewer's reports and your detailed responses to their comments (as Word file).
- 5) The paper explained: EMBO Molecular Medicine articles are accompanied by a summary of the articles to emphasize the major findings in the paper and their medical implications for the non-specialist reader. Please provide a draft summary of your article highlighting
 - the medical issue you are addressing,
 - the results obtained and
 - their clinical impact.This may be edited to ensure that readers understand the significance and context of the research. Please refer to any of our published articles for an example.
- 6) For more information: There is space at the end of each article to list relevant web links for further consultation by our readers. Could you identify some relevant ones and provide such information as well? Some examples are patient associations, relevant databases, OMIM/proteins/genes links, author's websites, etc...
- 7) Author contributions: the contribution of every author must be detailed in a separate section.
- 8) EMBO Molecular Medicine now requires a complete author checklist (<https://www.embopress.org/page/journal/17574684/authorguide>) to be submitted with all revised manuscripts. Please use the checklist as guideline for the sort of information we need WITHIN the manuscript. The checklist should only be filled with page numbers where the information can be found. This is particularly important for animal reporting, antibody dilutions (missing) and exact values and n that should be indicated instead of a range.

9) Every published paper now includes a 'Synopsis' to further enhance discoverability. Synopses are displayed on the journal webpage and are freely accessible to all readers. They include a short stand first (maximum of 300 characters, including space) as well as 2-5 one sentence bullet points that summarise the paper. Please write the bullet points to summarise the key NEW findings. They should be designed to be complementary to the abstract - i.e. not repeat the same text. We encourage inclusion of key acronyms and quantitative information (maximum of 30 words / bullet point). Please use the passive voice. Please attach these in a separate file or send them by email, we will incorporate them accordingly.

You are also welcome to suggest a striking image or visual abstract to illustrate your article. If you do please provide a jpeg file 550 px-wide x 300-800px high.

10) A Conflict of Interest statement should be provided in the main text

11) Please note that we now mandate that all corresponding authors list an ORCID digital identifier. This takes <90 seconds to complete. We encourage all authors to supply an ORCID identifier, which will be linked to their name for unambiguous name identification.

Currently, our records indicate that the ORCID for your account is 0000-0001-8858-5418.

Link Not Available

Photos 400-800 DPI

*Additional important information regarding figures and illustrations can be found at

<https://bit.ly/EMBOPressFigurePreparationGuideline>. See also figure legend preparation guidelines:

<https://www.embopress.org/page/journal/17574684/authorguide#figureformat>

***** Reviewer's comments *****

Referee #1 (Remarks for Author):

In the revised version, Sahu et al have demonstrated a commendable level of thoroughness in their work, particularly in characterizing the mechanistic aspects of protection upon reinfection. These experiments provide a comprehensive understanding of the mechanistic requirements of specific leukocyte subsets and effector molecules in CAN25-induced protection upon reinfection. There are, however, some additional points, as outlined below, which the authors could consider including in their Discussion.

1. In an attempt to unveil the cell types and key effector molecules that are critical for inducing protection, the authors either depleted the cells or neutralized the effector molecules using antibodies before secondary infection. This approach revealed that Gr-1+ PMNs/monocytes, CD4+ and CD8+ T cells, and IFN γ /IL17/TNF α are required for protection upon reinfection, in the vaccinated mice. While these data are solid, I believe that the question still remains as to how the vaccine strain mounts the protective response in the first place. In the Discussion, the authors should acknowledge these.

2. On Page 16, Lines 16 -18, the authors eloquently describe the roles of the cytokines. It would be valuable for the authors to also mention at this point that their data has significant implications, particularly in understanding the roles of these cytokines in vaccine-induced protection upon reinfection.

3. On Page 18, the authors discussed a potential trained immune response that may be elicited by the vaccine strain (Lines 3 - 13). While this is possible, the data do not conclusively support (or refute) the trained innate immune response in protective immunity. First, the T-cell subsets were found to be necessary for protection, which supports a role for the adaptive immune arm. Second, the effector molecules, such as IFN γ and IL-17 were also found to be critical for protection, implicating their functions in priming the myeloid phagocytes for protection. These are also in line with their observations with the SCID mice. Their conclusion from Gr-1 or clodronate-mediated depletion of the innate phagocytes and the resultant susceptibility clearly supports the function of these cells in protection upon reinfection; however, these data do not distinguish between the two possibilities of trained innate vs. adaptive immune-dependent myeloid priming upon reinfection. The authors should include a discussion of these two distinct possibilities which can be explored further in their next publications.

Referee #3 (Comments on Novelty/Model System for Author):

The authors have used several different approaches to understand the mechanism of a vaccine candidate they have created. The results are very interesting in that no vaccines for fungal diseases are available yet for medical use.

Referee #3 (Remarks for Author):

I would like to once again congratulate the authors on the significant effort they have put in this very interesting work. The authors have not accepted all of my suggestions, but I feel all of the point I raised have been adequately addressed.

Referee #1 (Remarks for Author):

In the revised version, Sahu et al have demonstrated a commendable level of thoroughness in their work, particularly in characterizing the mechanistic aspects of protection upon reinfection. These experiments provide a comprehensive understanding of the mechanistic requirements of specific leukocyte subsets and effector molecules in CAN25-induced protection upon reinfection. There are, however, some additional points, as outlined below, which the authors could consider including in their Discussion.

1. In an attempt to unveil the cell types and key effector molecules that are critical for inducing protection, the authors either depleted the cells or neutralized the effector molecules using antibodies before secondary infection. This approach revealed that Gr-1+ PMNs/monocytes, CD4+ and CD8+ T cells, and IFN γ /IL17/TNF α are required for protection upon reinfection, in the vaccinated mice. While these data are solid, I believe that the question still remains as to how the vaccine strain mounts the protective response in the first place. In the Discussion, the authors should acknowledge these.

Reply: Thank you so much for appreciating our efforts and agreeing with the provided mechanistic aspects of CNA25-induced protective immunity. We believe that the differential cell wall structure and composition between CNA25 and WT is the probable reason of a varied induction of various immune cell types and immune molecules in these fungal strains infected group and the reason for protective immune response in the first place. High content of β -glucan in CNA25 might have resulted in better recognition and immune activation than by the WT cells. It is well established that the Dectin-1- β -glucan interaction triggers the Card9-Syk pathway that leads to further activation of NF κ B and release of cytokines and chemokines. Dectin-1 also stimulates phagocytosis and inflammasome activation, however, activation such pathways need to be investigated further. Also a comparative proteomics analyses would shed light in the avirulence attributes of CNA25. It is now mentioned in the discussion portion (page 15).

2. On Page 16, Lines 16 -18, the authors eloquently describe the roles of the cytokines. It would be valuable for the authors to also mention at this point that their data has significant implications, particularly in understanding the roles of these cytokines in vaccine-induced protection upon reinfection.

Reply: Thank you so much, we have now included.

3. On Page 18, the authors discussed a potential trained immune response that may be elicited by the vaccine strain (Lines 3 - 13). While this is possible, the data do not conclusively support (or refute) the trained innate immune response in protective immunity. First, the T-cell subsets were found to be necessary for protection, which supports a role for the adaptive immune arm. Second, the effector molecules, such as IFN γ and IL-17 were also found to be critical for protection, implicating their functions in priming the myeloid phagocytes for protection. These are also in line with their observations with the SCID mice. Their conclusion from Gr-1 or clodronate-mediated depletion of the innate phagocytes and the resultant susceptibility clearly supports the function of these cells in protection upon reinfection; however, these data do not distinguish between the two possibilities of trained innate vs. adaptive immune-dependent myeloid priming upon reinfection. The authors should include a discussion of these two distinct possibilities which can be explored further in their next publications.

Reply: Thank you so much, we agree with you and a statement has now been included (in page 18).

Referee #3 (Comments on Novelty/Model System for Author and Remarks for Author):

The authors have used several different approaches to understand the mechanism of a vaccine candidate they have created. The results are very interesting in that no vaccines for fungal diseases are available yet for medical use.

I would like to once again congratulate the authors on the significant effort they have put in this very interesting work. The authors have not accepted all of my suggestions, but I feel all of the point I raised have been adequately addressed.

Reply: Thank you so much for appreciating and supporting our work.

Once again we thank the editors and reviewers for their valuable time spent reviewing our manuscript, we hope that the revised version will be acceptable for publication.

Sincerely,

Dr. Acharya

10th May 2024

Dear Dr. Acharya,

We are pleased to inform you that your manuscript is accepted for publication and is now being sent to our publisher to be included in the next available issue of EMBO Molecular Medicine.
